# Collision- and Reachability-Aware Multi-Robot Control with Grounded LLM Planners

## Abstract

Large language models (LLMs) have demonstrated strong performance in various robot control tasks. However, their deployment in real-world applications remains constrained. Even state-of-the-art LLMs, such as GPT-5, frequently produce invalid action plans that violate physical constraints, such as directing a robot to an unreachable location or causing collisions between robots. This issue primarily arises from a lack of awareness of these physical constraints during the reasoning process. To address this issue, we propose a novel framework that integrates reinforcement learning with verifiable rewards (RLVR) to incentivize knowledge of physical constraints into LLMs to induce constraints-aware reasoning during plan generation. In this approach, only valid action plans that successfully complete a control task receive positive rewards. We applied our method to two small-scale LLMs: a non-reasoning Qwen2.5-3B-Instruct and a reasoning Qwen3-4B. The experiment results demonstrate that constraint-aware small LLMs largely outperform large-scale models without constraint knowledge training, grounded on both the `BoxNet` task and a newly developed `BoxNet3D` environment built using MuJoCo, which involves LLM planning for up to 25 robots. This work highlights the effectiveness of grounding even small LLMs with physical constraints to enable scalable and efficient multi-robot control in complex, physically constrained environments. Our project website is at this link[1].

## 1 Introduction

Robotic control task requires controllers to find action plans given the robot's physical constraints. Conventional methods often employ planning tools, such as PDDL (Fox & Long, 2003) and temporal logics (Emerson, 1990) to find optimal plans. However, they often demand expert knowledge to convert task constraints to formal language and struggle to scale efficiently in multi-robot systems due to increased search time (Chen et al., 2024a; 2025; Huang et al., 2022). Recent advances in Large Language Models (LLMs), which excel at complex reasoning tasks like math and coding (Luo et al.; DeepSeek-AI, 2025; Shao et al., 2024; Liu & Zhang, 2025), have inspired their application in robotic control. LLMs can interpret natural-language task instructions and generate valid action plans (Meng et al., 2025; Chen et al., 2024b; Chu et al., 2025); for instance, ChatGPT can effectively generate high-level commands such as *"Robot A, move the square object to panel 2"* (Chen et al., 2024b; Mandi et al., 2023). Paired with low-level execution functions that translate these commands into control signals for robots, they have proven successful in various multi-robot tasks (Chen et al., 2024b; Mandi et al., 2023; Sun et al., 2022).

However, these successes have mainly been in synthetic or constrained environments, where physical interactions are overly simplified. For example, most tasks in RocoBench have predefined all the possible valid robot interactions with the objects, largely restricting the action space for LLMs. This has led to significant issues in real-world scenarios, where LLM planners tend to violate many basic physical constraints. In particular, two important constraints are often overlooked. **Reachability constraint:** LLM would direct a robot arm to an unreachable position (Chen et al., 2024b; Zhang et al., 2025). **Collision constraint:** LLM would schedule robots to the same space, leading to collisions (Mandi et al., 2023; Jones et al., 2025).

---

[1] https://anonym-submission-user.github.io

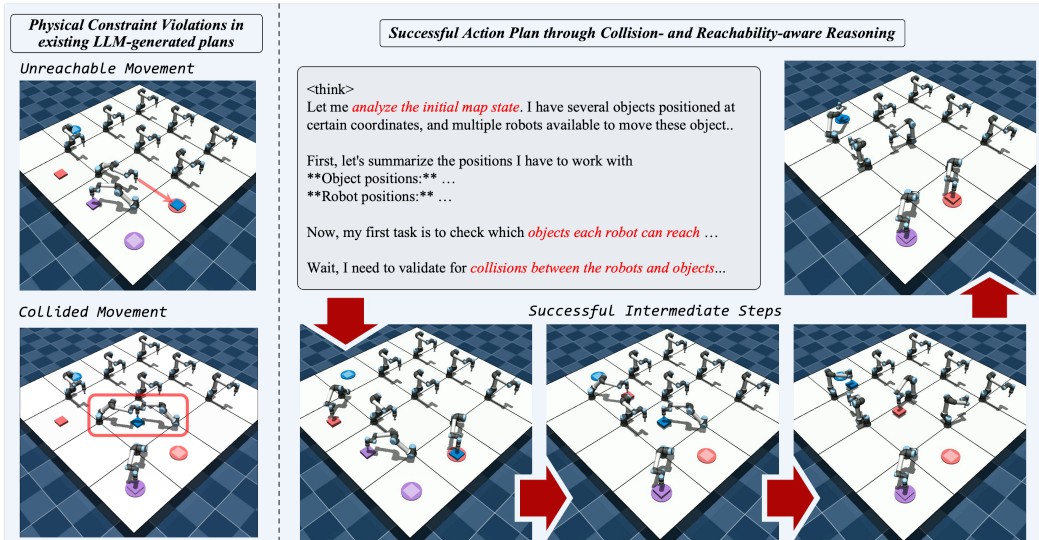

Figure 1: Illustration of LLM-based multi-robot control. (Left) Without grounding constraint knowledge, the LLM generates action plans that result in unreachable positions or collisions. (Right) Our planner generates valid movement actions through constraint-aware reasoning (highlighted in red) that successfully completes the `BoxNet` task after grounding robotic constraints knowledge.

As an example, Figure 1 (left) shows invalid actions generated by a SOTA reasoning LLM GPT-5, which easily violate these constraints, leading to significant safety and feasibility concerns.

These issues highlight the imperative to equip LLM planners with the ability to understand, analyze, and adhere to basic physical constraints. However, incorporating these constraints would require strong geometric reasoning and self-reflection capabilities, particularly when the number of robots is large, which may pose nontrivial challenges to LLMs. This raises a key research question: *Can LLMs, given their current reasoning capabilities, be trained to integrate physical constraints into the planning process? If so, to what extent can they succeed?*

To study these research questions, this paper presents a novel framework to incentivize these physical constraints into LLM planners, enabling them to reason about action validity during plan generation. Specifically, we leverage reinforcement learning with verifiable rewards (RLVR) that incorporates checks for reachability, kinematic feasibility, and collision avoidance. By using binary success/-failure signals derived from the robot control environment, we ensure that the LLM only receives rewards for generating physically valid plans. This fine-tuning process enables the LLM to reason about the validity during plan generation, leading to more reliable and collision-free action plans.

Our experiments on two LLMs, a non-reasoning Qwen2.5-3B-Instruct and a reasoning Qwen3-4B, have shown several encouraging findings. First, by incorporating the physical constraints into the reward, LLM planners can quickly acquire the ability to adhere to the physical constraints, thus drastically increasing the planning success rate, outperforming SOTA large-scale LLMs. For example, our best planner can achieve 0.87 and 0.53 pass@1 on two `BoxNet`-task multi-robot datasets that require controlling up to 25 robots, while the best baseline planner can only achieve 0.52 and 0.39 pass@1, respectively. Figure 1 (right) shows the thinking process and the generated plan by our fine-tuned LLM, which successfully solves the task without violating physical constraints. Second, our reasoning probing experiments have revealed that LLMs indeed learn to correctly identify whether the geometric constraints are satisfied or not. Finally, such capabilities acquired from RL can generalize to unseen environments with new environment size and geometric configurations, which further verifies that LLMs learn the generic reasoning skills rather than overfitting to specific geometric configurations.

In summary, the contributions of this work are as follows:

- We propose a novel framework that grounds LLMs with knowledge of action validity and collision constraints, ensuring LLM-planner-generated plans avoid unreachable positions, object collisions, or robot collisions.

- We introduce two new environments based on `BoxNet` task featuring tasks with up to 25 and 9 robots, respectively, which incorporate realistic physical constraints and serve as testbeds for evaluating LLM-based multi-robot control.

- We implement our approach on two small-scale LLMs, demonstrating that even small models like Qwen2.5-3B-Instruct and Qwen3-4B, when grounded with physical constraints, can outperform larger, state-of-the-art LLMs with no internal physical constraint knowledge in complex multi-robot control tasks.

## 2  METHOD

### 2.1  OVERVIEW

In this section, we introduce our framework for grounding LLMs with reachability and collision awareness. Denote $\mathcal{M}_{\boldsymbol{\theta}_0}(\cdot)$ as the initial LLM for performing the robot control tasks, which is capable of generating an action plan $\boldsymbol{s} \sim \mathcal{M}_{\boldsymbol{\theta}_0}(\boldsymbol{q}; \mathcal{C})$ for solving the given control task described by $\boldsymbol{q}$ under a set of physical constraints $\mathcal{C}$, such as the reachability of a robot arm and collision avoidance. Our goal is to fine-tune the LLM such that the generated solution $\boldsymbol{s}$ successfully moves objects to their target positions while not violating the constraints $\mathcal{C}$. In the following, we first introduce the RLVR framework for grounding physical constraints in Section 2.2, then introduce our initial LLM policy warmup strategy in Section 2.3, and two different planner modes we consider in Section 2.4.

### 2.2  GROUNDING LLM WITH PHYSICAL CONSTRAINTS THROUGH RLVR

We adopt a similar RL framework to the DeepSeek-R1 LLM (DeepSeek-AI, 2025; Ren et al., 2025), which employs the group relative policy optimization (GRPO) algorithm (Shao et al., 2024). Specifically, at each training step $i, i \geq 1$, we sample a group of different plans and its corresponding reasoning $\{\boldsymbol{s}_1, \boldsymbol{s}_2, \ldots, \boldsymbol{s}_G\}$ from the old LLM policy $\mathcal{M}_{\boldsymbol{\theta}_{i-1}}$ for each query robot-control task $\boldsymbol{q}$, where $G$ is the group size. Each plan $\boldsymbol{s}_j$ is simulated in a manually implemented environment with physical constraints. The corresponding reward function $r(\cdot)$ later estimates whether it successfully completes the given task with the simulation environment feedback. Then the LLM is optimized by maximizing the following objective (Shao et al., 2024).

$$
\mathcal{J}_{GRPO}(\mathcal{M}_{\boldsymbol{\theta}_i}) = \mathbb{E}\left[\left(\boldsymbol{q} \sim \mathcal{D}, \{\boldsymbol{s}_j\}_{j=1}^G \sim \mathcal{M}_{\boldsymbol{\theta}_{i-1}}(\boldsymbol{O}|\boldsymbol{q}; \mathcal{C})\right)\right]
$$

$$
= \frac{1}{G}\sum_{j=1}^G \left(\min\left(\frac{\mathcal{M}_\theta(\boldsymbol{s}_j|\boldsymbol{q}; \mathcal{C}))}{\mathcal{M}_{\boldsymbol{\theta}_{i-1}}(\boldsymbol{s}_j|\boldsymbol{q}; \mathcal{C}))}A_j, \mathrm{clip}\left(\frac{\mathcal{M}_\theta(\boldsymbol{s}_j|\boldsymbol{q}; \mathcal{C}))}{\mathcal{M}_{\boldsymbol{\theta}_{i-1}}(\boldsymbol{s}_j|\boldsymbol{q}; \mathcal{C}))}, 1-\epsilon, 1+\epsilon\right)A_j\right)
$$

$$
- \beta \mathbb{D}_{KL}\left(\mathcal{M}_{\boldsymbol{\theta}_i} \| \mathcal{M}_{\boldsymbol{\theta}_0}\right),
$$

where $\mathcal{D}$ denotes the training data and $A_j$ represents the advantage, computed as the reward of each plan subtracted by the average reward within the group. Detailed definitions are in Appendix D.

Our reward function, denoted as $r(\cdot)$ largely follows the design in DeepSeek-R1 (DeepSeek-AI, 2025), with an additional plan efficiency term. Specifically,

$$
r(\boldsymbol{s}; \boldsymbol{q}, \boldsymbol{s}^*, \mathcal{C}) = r_{\text{format}}(\boldsymbol{s}) + r_{\text{execute}}(\boldsymbol{s}; \mathcal{C}) - r_{\text{efficiency}}(\boldsymbol{s}; \boldsymbol{s}^*),
$$

$r_{\text{format}}(\boldsymbol{s}) = 0.1$ if the generated solution adheres to the required thinking-then-response format and $0$ otherwise. $r_{\text{execute}}(\boldsymbol{s}; \mathcal{C}) = 1$ if the simulator verifies that the plan ❶ accomplishes the task AND ❷ no physical constraints are violated, and $0$ otherwise. Incorporating physical constraint checking in $r_{\text{execute}}$ is the key mechanism to improve constraint awareness of the LLM planner. Finally,

$$
r_{\text{efficiency}}(\boldsymbol{s}; \boldsymbol{s}^*) = \max\left(0, \ 0.1 \times (\mathrm{len}(\boldsymbol{s}) - \mathrm{len}(\boldsymbol{s}^*))\right)
$$

penalizes the excessive length compared to a golden plan, $\boldsymbol{s}^*$, which is obtained by an A* search algorithm (see Appendix D for search details). Additionally, we implement a minimum cap of $2 \times r_{\text{format}}(\boldsymbol{s})$ when $\boldsymbol{s}$ is a valid plan that successfully solves the task to ensure that correct plans always receive a higher reward than incorrect ones.

Table 1: Example synthesized reasoning trace for FULLPLAN planner and REPLAN planner. We highlight three reasoning behaviors: 1. environment analysis in blue, 2. validity verification in green, and 3. efficiency consideration in orange. The environment feedback is marked in gray.

| **FULLPLAN Planner Reasoning**: | **REPLAN Planner Reasoning**: |
|---|---|
| Object positions: | <observation> |
|     Object 0: [0.75, 1.75] . . . | Object positions: |
| <think>Okay, let me analyze the given environment before coming up with a movement plan . . . . |     Object 0: [0.75, 2.75] . . . |
| ### Evaluating Reachability: | </observation> |
| - **Robot 0** cannot reach either object . . . | <think>Alright, let's break down the situation . . . |
| ### Collision Checks: | <observation> |
| - No objects are placed at the same position. . . | Object positions: |
| Let me finalize this, since there are no collisions, this plan efficiently moves all objects . . . |     Object 1: [2.25, 2.75] . . . |
|  | </observation> |
|  | <think>Alright, let's analyze this . . . |

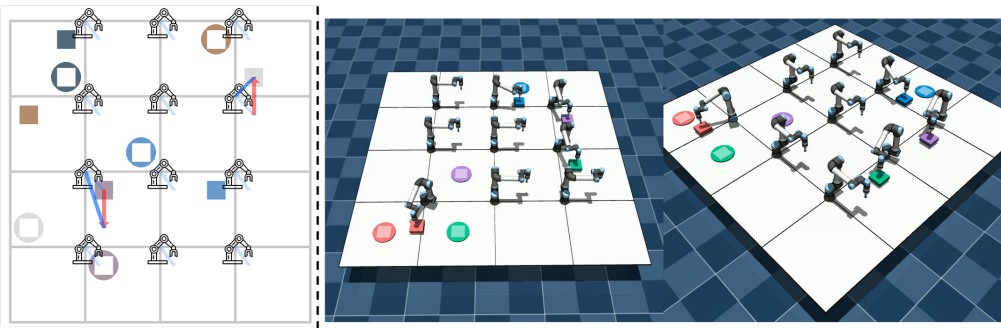

Figure 2: (Left) An example `BoxNet2D` environment. The blue lines mark the robot arm, and the red lines mark the movement trajectory. (Right) An example `BoxNet3D` environment. Both environments require robots to collaborate to move boxes to the circle with the corresponding color.

## 2.3 INITIAL SUPERVISED FINE-TUNING (SFT) WARMUP

Prior works have shown that LLMs' initial performance on a reasoning task is crucial to RLVR training (Luo et al.; Shao et al., 2024; Liu & Zhang, 2025). Since off-the-shelf LLMs often struggle with robotic control tasks, we introduce an SFT warmup to equip them with basic robot control knowledge before RL training.

The SFT data needs to contain two components: ❶ a correct plan to solve a given task, and ❷ a reasoning chain that reflects a multi-step decision-making process leading to the correct plan. To synthesize such data, for each task, we first use the A* search algorithm to search for the optimal plan. Then, we pass the plan to an LLM, which is prompted to generate a reasoning process for the plan, consisting of the following three patterns:

- *Analysis of the given environment*, where the LLM assesses the current positions of robot arms and objects, *e.g.*, *let me analyze the current situation*;
- *Validity verification*, where the LLM reasons about an arm's reachable area based on its base position and potential collision between different arms, *e.g.*, *If Robot 0 moves, . . . , it will collide*;
- *Efficiency considerations*, where the LLM evaluates whether multiple movements can be parallelized to improve the plan efficiency.

Table 1 shows the example reasoning chain synthesized by GPT4o-mini, where the three patterns are rendered in different colors. Appendix F shows the full prompt for our reasoning synthesis.

## 2.4 TWO PLANNERS: FULLPLAN PLANNER AND REPLAN PLANNER

We consider two different LLM-based planners in this work. The first planner, referred to as FULLPLAN, involves the LLM directly generating the entire plan that may take multiple execution steps for solving a task based on the initial positions of all objects and robot poses in the environment. The second planner, denoted as REPLAN, generates one step at a time, observing the updated object positions from the environment (appended to its context) before generating the next

step. This allows the planner to evolve dynamically as the environment changes through multiple steps. Table 1 provides the example planning processes for two different LLM-based planners for the same initial environment. We highlight that the REPLAN planner observes multiple intermediate observations of the environment, while the FULLPLAN planner only sees the initial environment.

## 3 BOXNET-BASED MULTI-ROBOT ENVIRONMENTS

In this work, we primarily experiment with `BoxNet` task (Chen et al., 2024b), where multiple robots collaborate to move objects across different cells to targeted locations in a fixed grid map. This section details two environments we developed, a modified `BoxNet2D` environment and the newly developed `BoxNet3D` environment, both equipped with realistic physical constraint checks.

**BoxNet2D.** Figure 2 (left) shows a `BoxNet2D` environment. In this environment, robot arms are placed at a corner of a grid environment. Each arm can reach its neighboring grids for picking and placing objects. Unlike the previous `BoxNet` environment that predefined all valid robot arm actions, we allow LLMs to generate spatial coordinates directly, significantly expanding the action space. For example, the action *"Robot0 Move (1.25, 1.25) → (1.75, 1.75), False"* moves *Robot0*'s arm to *(1.75, 1.75)* without picking up an object. In contrast, *"Robot1 Move (2.25, 1.75) → (1.25, 1.25), True"* indicates *Robot1* picking up the object at *(1.75, 1.25)* and moving it to *(2.25, 1.75)*. For `BoxNet2D` environment, each arm moves along a straight path from the start point to target point.

We pre-define four points within each grid, *e.g.*, *(0.25, 0.25), (0.25, 0.75), (0.75, 0.25), (0.75,0.75)*, for object placement and robot arm moving, and later we will show that the fine-tuned LLM can generalize to other points in experiments. Three physical constraints are incorporated: ❶ *reachability verification*, which checks whether the target position of a robot is unreachable. ❷ *robot collision detection*, which checks whether the movement trajectory of different arms intersects with each other, or one robot's movement trajectory intersects with another robot arm, leading to a collision. ❸ *object collision detection*, which checks whether two objects are placed at the same spatial coordinates during the plan execution. To implement these constraints, we manually implement these constraint detection functions and raises error when any constraint is not satisfied. For example, reachability verification checks whether an arm's position has larger offset to its base, *i.e.*, $\Delta x > 1$ or $\Delta y$. The robot collision constraint detection whether two arm movement paths has intersecting point. Example invalid actions of `BoxNet2D` are provided in Appendix E.

**BoxNet3D.** Figure 2 (right) shows a `BoxNet3D` environment. In this environment, we employ the UR5e robot arm as the basic robot arm[2]. Similar to the 2D environment, the goal is to move colored boxes into corresponding circles of the same color with the fewest actions. Each robot arm's base is fixed at grid joint and moves its arm around to reach adjacent grid center position for object pickup and placement. We employ an RRT planner implemented by RoCoBench (Mandi et al., 2023) for low-level control signal generation, *i.e.*, the robot joint configuration trajectory during each step of movement, given LLM planner-generated coordinates for arm position movement[3]. We employ Mujoco as the engine (Todorov et al., 2012) to power arm reachability check and collision detection, where unreachable position result in failed robot joint calculation in the low-level RRT planner, and collision result in geometry contact that can be captured by Mujoco engine. Example invalid actions of `BoxNet3D` are provided in Appendix E.

## 4 EXPERIMENT

In this section, we conduct empirical experiments on the two `BoxNet`-based environments to assess the effectiveness of our method. We first present the experiment setup in Section 4.1 and then the experiment results in Section 4.2, followed by additional ablation studies in Section 4.3.

---

[2]https://www.universal-robots.com/products/ur5e/

[3]The RRT planner is adapted from the implementation in RoCoBench official code base (https://github.com/MandiZhao/robot-collab/blob/main/rocobench/rrt.py)

Table 2: Performance of different LLM planners on `BoxNet2D` and `BoxNet3D`. For each task, we report *Success*, ratio of pass@1 accuracy over four trials, *StepDiff*, the difference in number of steps between model-generated plans and A* searched plans for successful executions, and *Para.*, the maximum number of robots operatin in parallel. For each model, the performance for FULLPLAN planner and REPLAN planner side-by-side (FULLPLAN / REPLAN).

| Model | BoxNet2D | | | BoxNet3D | | |
| | Success ↑ | StepDiff. ↓ | Para. ↑ | Success ↑ | StepDiff. ↓ | Para. ↑ |
| --- | --- | --- | --- | --- | --- | --- |
| Search Algorithm | | | | | | |
| A* | 1 | 0 | 2.24 | 1 | 0 | 2.14 |
| LLMs without constraint knowledge grounding | | | | | | |
| GPT-4omini | 0.06 / 0.05 | 2.35 / 0.14 | 1.17 / 1.15 | 0.07 / 0.06 | 0.79 / 0.45 | 1.03 / 1.08 |
| GPT-4o | 0.12 / 0.11 | 2.14 / 0.13 | 1.15 / 1.22 | 0.10 / 0.12 | 0.23 / 0.68 | 1.35 / 1.11 |
| GPT-o4mini | 0.37 / 0.35 | 0.24 / -0.31 | 1.58 / 1.87 | 0.11 / 0.33 | 0.14 / 1.21 | **1.45** / 1.53 |
| GPT-5-mini | 0.43 / 0.31 | 1.13 / 0.44 | 1.44 / 1.75 | 0.23 / 0.30 | 0.25 / 0.44 | 1.24 / 1.33 |
| GPT-5 | 0.52 / 0.44 | 0.78 / 0.13 | 1.52 / 1.61 | 0.31 / 0.39 | 0.29 / 0.37 | 1.34 / 1.21 |
| Qwen2.5-3B-Inst | 0.0 / 0.0 | —— / —— | —— / —— | 0.08 / 0.0 | 0.20 / —— | 1.40 / —— |
| Qwen2.5-7B-Inst | 0.02 / 0.02 | 1.45 / 0.31 | 1.20 / 1.23 | 0.05 / 0.08 | 0.41 / 0.35 | 1.13 / 1.07 |
| QwQ-32B | 0.04 / 0.07 | 0.35 / 0.17 | 1.12 / 1.21 | 0.07 / 0.15 | 0.24 / -0.09 | 1.08 / 1.31 |
| Qwen3-4B | 0.14 / 0.13 | 0.23 / 0.14 | 1.29 / 1.29 | 0.15 / 0.11 | 0.07 / 0.31 | 1.17 / 1.14 |
| Qwen3-8B | 0.18 / 0.15 | -0.23 / -0.34 | 1.24 / 1.31 | 0.17 / 0.13 | -0.02 / 0.09 | 1.22 / 1.21 |
| Qwen3-14B | 0.19 / 0.21 | -0.31 / -0.24 | 1.34 / 1.41 | 0.10 / 0.14 | 0.17 / 1.37 | 1.34 / 1.35 |
| Qwen3-32B | 0.11 / 0.14 | 0.17 / -0.03 | 1.24 / 1.12 | 0.14 / 0.17 | 0.09 / 0.04 | 1.27 / 1.09 |
| LLMs with grounded constraint knowledge | | | | | | |
| Qwen2.5-3B-SFT | 0.34 / 0.30 | 0.11 / -0.04 | 1.51 / 1.39 | 0.27 / 0.39 | -0.07 / -0.05 | 1.27 / 1.39 |
| Qwen2.5-3B-RL | 0.58 / 0.68 | -0.65 / 0.23 | 1.53 / 1.50 | 0.42 / 0.48 | -0.15 / -0.14 | 1.33 / 1.49 |
| Qwen3-4B-SFT | 0.45 / 0.31 | -0.12 / -0.15 | **1.92** / 1.35 | 0.37 / 0.43 | 0.09 / -0.11 | 1.32 / 1.48 |
| Qwen3-4B-RL | **0.87** / **0.75** | **-0.84** / **-0.64** | 1.73 / **1.64** | **0.45** / **0.53** | **-0.25** / **-0.29** | 1.39 / **1.56** |

## 4.1 EXPERIMENT SETUP

**Dataset generation.** We create environments with various map sizes and object initial and target positions for both `BoxNet2D` and `BoxNet3D` . Specifically, for `BoxNet2D` , we use map sizes ranging from $2 \times 2$ to $6 \times 6$ and 1 to 5 objects, resulting in 55,000 training and 250 testing environments. For `BoxNet3D` , we use map sizes from $2 \times 2$ to $4 \times 4$ with 1 to 3 objects, yielding 1,800 training and 160 testing environments. The object position is randomly sampled from the pre-defined points, while the robot arms are evenly placed at the grid joints to ensure that all grids in the map can be reached. With the robot placement strategy introduced earlier, the maximum number of robots reaches up to 25 and 9 for `BoxNet2D` and `BoxNet3D`, respectively. For each randomly sampled environment, the manually implemented A* search algorithm verifies that a valid solution action plan exists. While our focus is on solvable environments, we include a discussion of unsolvable cases in Appendix C.4. Detailed dataset statistics are summarized in Appendix D.3.

**Evaluation metric.** We evaluate LLM-based planners mainly from two perspectives: ❶ *Success*, the proportion of generated plans that solve given robotic tasks, measured by *pass@1* over four trials per environment; and ❷ *StepDiff.*, the difference in number of steps between successful plans and the best plan among A* solutions. We also report *Para.*, the maximum number of robots operating in parallel in any intermediate step of a successful plan.

**Baseline LLMs.** We mainly compare with off-the-shelf LLMs via direct prompting on robot control task. To ensure comprehensive coverage of existing LLMs, our evaluation includes both reasoning and non-reasoning models, closed-source and open-source ones across different parameter scales. Specifically, we consider closed-source LLMs, GPT-4o, GPT-4o-mini, GPT-o4-mini, GPT-5-mini and GPT-5. On the open-source side, we include Qwen-2.5 and Qwen3 series, with parameter sizes ranging from 3B to 32B. We also compare with LLM-based symbolic translation approach such as AutoTAMP (Chen et al., 2024a) in Appendix C.1.

**Training details.** We use two base LLMs: a non-reasoning LLM Qwen-2.5-3B-Instruct and a reasoning LLM Qwen3-4B. For SFT warm-up, we use a learning rate of $1e-5$ for Qwen-2.5-3B-Instruct and $3e-5$ for Qwen-3-4B with the AdamW optimizer (Loshchilov & Hutter, 2017). Training runs for 10 epochs on FULLPLAN and 5 epochs on REPLAN . RLVR training uses a fixed $1e-6$ learning rate for 200 steps with the GRPO algorithm (Guo et al., 2025). Batch sizes are 256 (group size 8) for `BoxNet2D` and 64 for `BoxNet3D` . Following prior work (Luo et al.; Liu & Zhang, 2025), we set $\beta = 0$ in the GRPO loss. We use the VeRL framework (Sheng et al., 2024), and run all experiments on $2 \times 8$ NVIDIA H100 GPUs. Detailed computation cost is in Appendix D.2.

## 4.2 EXPERIMENTAL RESULTS

**Grounding empowers small-scale LLMs to outperform larger ones.** We first evaluate the grounded LLM planner performance in Table 2 and a detailed error type breakdown is in Appendix C.3. The LLMs with physical constraints knowledge grounded through SFT warmup and further RL training are denoted by the suffix *-SFT* and *-RL*, respectively.

We highlight three observations: First, grounding constraint knowledge significantly boosts planning performance, enabling 3B and 4B LLMs to achieve higher success against larger ones. For example, Qwen3-4B-RL FULLPLAN planner achieves 0.87 success rate, 0.5 higher than the best baseline GPT-o4mini. Second, grounded LLM planners produce more efficient plans than the A* search algorithm on solved tasks. For example, Qwen3-4B-RL has 0.84 fewer steps than the ground-truth plan from our A* implementation, showing a strong reasoning ability and also echoes the findings in prior works that compare LLM planners with A* on Sudoku (Lehnert et al., 2024; Su et al., 2024). Third, planner performance on `BoxNet3D` is generally worse than on `BoxNet2D`. This suggests that, although we applied multiple feasibility checks in `BoxNet2D`, some physical constraints remain missing. The `BoxNet3D` environment uses a more advanced simulation engine and thus exposes more limi-

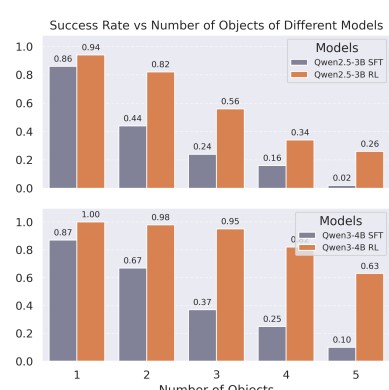

Figure 3: Success rate against number of boxes (up) and against in the `BoxNet2D` environment.

tations. These results underscore the importance of developing realistic robotic environments that capture real-world complexity for future LLM-based robotic control research.

Figure 3 visualizes performance against the number of boxes for `BoxNet2D`. We highlight that RL-trained planners better preserve performance when task complexity increases. For example, the performance gap between Qwen3-4B-SFT and Qwen3-4B-RL grows from 0.13 to 0.53 when the number of boxes increases from 1 to 5, highlighting better scalability of RL planners.

**RL planners generalize better to unseen environments.** To measure how the planners' reasoning ability generalizes, we evaluate the planners' performance in two unseen variants of `BoxNet2D` test data: ❶ Random robot layout, denoted as RANDROB, where the robot positions are randomly assigned on the grid joints in maps ranging from $2 \times 2$ to $5 \times 5$. We ensure that all testing data are solvable, which means every box

Table 3: Planning performance generalization on unseen `BoxNet2D` environments.

| Model | RANDROB | | NEWCOORD | | UNSEENMAP | |
|---|---|---|---|---|---|---|
| | Success ↑ | StepDiff. ↓ | Success ↑ | StepDiff. ↓ | Success ↑ | StepDiff. ↓ |
| FULLPLAN Planner | | | | | | |
| Qwen2.5-3B-SFT | 0.39 | 0.12 | 0.32 | 0.21 | 0.25 | 0.17 |
| Qwen2.5-3B-RL | 0.58 | -0.04 | 0.55 | -0.32 | 0.35 | 0.05 |
| Qwen3-4B-SFT | 0.48 | 0.03 | 0.43 | -0.03 | 0.30 | 0.11 |
| Qwen3-4B-RL | **0.79** | **-0.40** | **0.87** | **-0.39** | **0.40** | **-0.24** |
| REPLAN Planner | | | | | | |
| Qwen2.5-3B-SFT | 0.39 | 0.23 | 0.33 | **0.09** | 0.18 | 0.22 |
| Qwen2.5-3B-RL | 0.71 | 1.24 | 0.68 | 1.04 | 0.39 | 0.14 |
| Qwen3-4B-SFT | 0.41 | -0.03 | 0.37 | 0.15 | 0.32 | -0.05 |
| Qwen3-4B-RL | **0.75** | **-0.15** | **0.69** | 0.09 | **0.42** | **-0.11** |

can reach its target position via robot movement. ❷ Unseen coordinates, *i.e.*, NEWCOORD, where the initial and target position coordinates of all objects in `BoxNet2D` test set are perturbed by a random offset $(\Delta x, \Delta y) \sim \mathcal{U}([-0.2, 0.2]^2)$. ❸ Unseen map size, *i.e.*, NEWMAP, where the testing grid size is extended to two unseen sizes $10 \times 5$ and $7 \times 7$. Example data are shown in Appendix E.

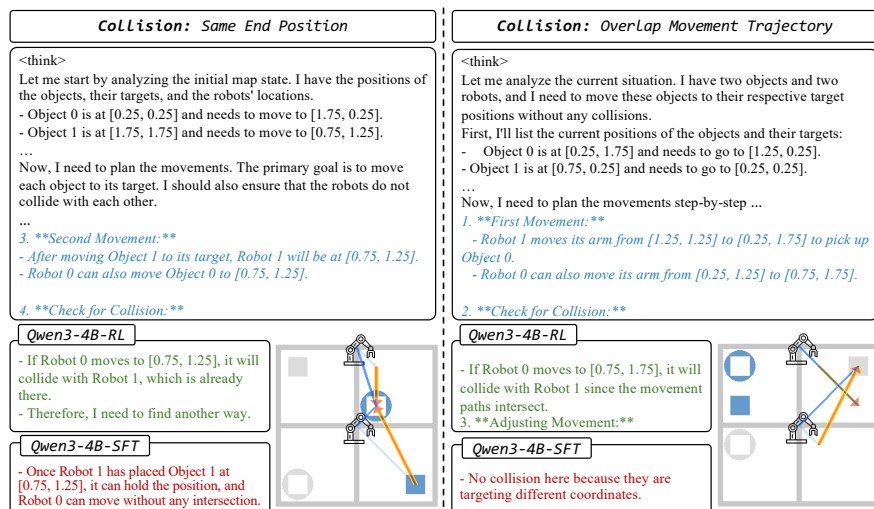

Figure 4: Example reasoning trace generated by grounded FULLPLAN planners. Invalid action plans are manually inserted into the trace history and highlighted in blue. The correct continuations that identify and correct these errors are shown in green, while incorrect continuations are shown in red. The bottom figure visualizes the collision between two movements. RL planner better detects errors.

Table 3 reports the performance of our grounded planners on three unseen environments. We highlight that the RL-trained planners consistently outperform SFT ones while maintaining plan efficiency. For example, Qwen3-4B-RL FULLPLAN planner achieves 0.87 success rate on NEWCOORD, 0.44 better than the SFT variant, showing better generalization of the reasoning capability. This observation also aligns with previous RL for LLM works (Chu et al., 2025; Razin et al., 2025).

**Reasoning behavior change after RL.** Previous results have shown that RL training significantly improves the planner's planning ability. In this section, we analyze in more detail how the reasoning behavior of the LLM-based planners changes before and after RL fine-tuning. Given the critical role of reachability checks and collision checks in generating successful action plans, we prompt GPT-4o to count the number of these checks in the reasoning traces produced by our FULLPLAN planners across three `BoxNet2D` environment variants. The prompts used are provided in Appendix F.

As shown in Table 4, the RL planners perform more reachability checks (Rea.) and collision checks (Col.) than the initial SFT planner. These checks help ensure the feasibility of action plans and lead to a large improvement in success rate. This observation suggests that RL training helps the LLM better understand the importance of these checks and use them more consistently.

Table 4: Number of different reasoning behaviors for grounded FULLPLAN planners.

| Model | BoxNet2D Rea. | BoxNet2D Col. | RANDROB Rea. | RANDROB Col. | NEWCOORD Rea. | NEWCOORD Col. |
|---|---|---|---|---|---|---|
| Qwen2.5-3B-SFT | 8.0 | 9.8 | 8.4 | 8.3 | 8.9 | 9.1 |
| Qwen2.5-3B-RL | 8.6 | 10.4 | 9.2 | 8.9 | 9.8 | 9.5 |
| Qwen3-4B-SFT | 9.1 | 7.4 | 7.3 | 6.4 | 10.1 | 9.4 |
| Qwen3-4B-RL | 10.1 | 8.2 | 7.7 | 6.7 | 10.3 | 9.6 |

We also verify their reasoning ability to identify and recover from intermediate errors by manually injecting error steps into the trace. Specifically, we insert an invalid action that would lead to a collision into the intermediate reasoning steps. To test whether the planner can recognize and correct such errors, we append the phrase *"Collision Check"* to the perturbed trace to trigger verification. Figure 4 shows two qualitative examples, where the injected invalid actions are highlighted in blue. The LLM's continuation is marked in green if it identifies and corrects the error, and in red if it fails. We find that the RL-trained planner successfully finds the error and traces the issue to same target position and overlapped movement paths. This suggests that RL helps build better physical constraints-aware reasoning. Further quantitative analysis results are presented in Appendix C.2.

## 4.3 ABLATION STUDY

In this section, we explore the design choices for our framework on the `BoxNet2D` environment, focusing on: ❶ How does SFT warmup affect final planner performance? ❷ Is the textual thinking necessary for planners? ❸ How does the efficiency penalty affect the planner's behavior?

**Initial LLM policy matters.** Figure 5 shows how the training reward evolves over the first 40 steps for different initial FULLPLAN planners. For the original Qwen2.5-3 B-Instruct and Qwen3-4B models, we observe a sharp reward increase of about 0.1 (the format reward) within the first 10 steps, which indicates that they quickly learn to produce answers in required format. This observation aligns with previous RLVR works (DeepSeek-AI, 2025) However, after this initial gain, the reward plateaus, indicating limited additional learning to further improve the LLM's planning capability.

In contrast, the LLM with SFT warmup shows a consistent increasing reward trend. This suggests that the SFT warmup helps build a strong foundation, allowing it to continue learning and optimize effectively in RL.

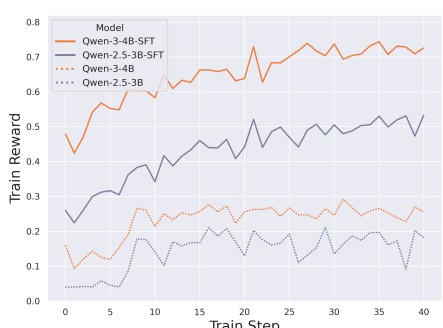

**Textual reasoning improves planning performance.** To assess the role of textual reasoning for LLM planners, we perform an ablation study with our SFT and RL pipeline on Qwen-2.5-3B-Instruct. In this experiment, we train a planner without the synthesized reasoning, *i.e.*, it generates only the final action plan with no textual thinking. As shown in Table 5, removing intermediate reasoning leads to a notable performance drop: success rates fall from 0.34 to 0.26 for SFT planner, and from 0.58 to 0.39 for RL planner. This shows the importance of textual thinking for LLM planners.

Figure 5: Training reward trajectory in first 40 steps for different initial LLMs.

**Efficiency penalty in reward improves plan efficiency.** We observed a surprising finding that RL-trained LLM planners produce more efficient plans than those generated by our hand-crafted A* search algorithm in Table 2, which is likely due to the efficiency penalty term in the reward function. To further understand its role, we conduct an ablation study on $r_{\text{efficiency}}$. Starting from the same initial LLM policy Qwen-2.5-3B-SFT, we perform RL training without the efficiency penalty for FULLPLAN planner. Table 5 presents the results.

Table 5: Impact of ablating thinking and $r_{\text{efficiency}}$ on `BoxNet2D` performance.

| Model | BoxNet2D | | |
|---|---|---|---|
| | Success ↑ | StepDiff. ↓ | Para. ↑ |
| Qwen2.5-SFT | 0.34 | 0.11 | 1.51 |
| − thinking | 0.26 | 0.05 | 1.35 |
| Qwen2.5-RL | **0.58** | **-0.65** | **1.53** |
| − thinking | 0.39 | -0.07 | 1.43 |
| − $r_{\text{efficiency}}$ | 0.52 | 1.44 | 1.07 |

While both RL-finetuned LLM largely improve the success rate, their plan efficiency differs significantly. The planner trained with $r_{\text{efficiency}}$ produces plans that are 2.09 steps shorter. In contrast, the parallelism drops close to 1 without the penalty in RL. These results underscore the importance of efficiency penalty in reward.

## 5 RELATED WORK

**Robotic planning and control with LLMs.** Classical robot task planning involves formalizing goals and physical constraints with Temporal Logic or PDDL and solving them with constraint solvers (Fox & Long, 2003; Emerson, 1990). LLM-based alternatives either select among motion primitives (Guan et al., 2023; Skreta et al., 2023; Loula et al., 2025), generate code as a control interface (Chen et al., 2025; Huang et al., 2022; Meng et al., 2025; Liang et al., 2023; Ahn et al., 2022; Singh et al., 2023), combine with classical planners (Chen et al., 2024a; Lin et al., 2023), or use multi-LLM discussion (Chen et al., 2024b; Zhang et al., 2024; Guo et al., 2024; Shen et al., 2025b). Despite promising progress, many methods overly simplify physical constraints; we show that even SOTA LLMs struggle under realistic constraints and introduce a method that grounds smaller LLMs with constraint knowledge to substantially improve performance.

**Reinforcement learning with verifiable rewards for LLM reasoning.** RL with verifiable rewards has strengthened LLM reasoning in mathematics (Ren et al., 2025; Guo et al., 2025; Zeng et al., 2025), code (Liu & Zhang, 2025; Luo et al.; OpenAI, 2025), and multi-agent settings (Jin et al., 2025; Singh et al., 2025; Feng et al., 2024; OpenAI, 2024). Such training often induces emergent behaviors like feasibility checks and self-reflection, beyond supervised fine-tuning (Chu et al., 2025; Zelikman et al., 2024; Pan et al., 2025; Shen et al., 2025a; Hou et al., 2025). We extend RLVR to robotic control by integrating physical verification signals (collision avoidance, reachability, goal satisfaction) into training, thereby grounding constraint knowledge and yielding more robust plans. Additional related work discussion is provided in Appendix B.

## 6 CONCLUSION

In this paper, we present a novel framework that grounds LLMs with physical constraint knowledge, such as robot arm reachability and collision avoidance. By incorporating these constraints, LLMs are able to reason more effectively about action feasibility and generate efficient and physically viable action plans. To evaluate our approach, we developed two `BoxNet`-based multi-robot environments, `BoxNet2D` and `BoxNet3D`, both equipped with action feasibility checks. Experiments show that even small-scale LLMs at 3B and 4B parameter size, when grounded with constraint knowledge, significantly outperform larger SOTA LLMs without constraint knowledge training. Additional experiments on reasoning behavior and generalization further confirm that our models learn constraint-aware reasoning rather than simply overfitting to training data.

**Ethic Statement** This work aims to enhance the LLM-based multi-robot controller with an enhanced training pipeline to incorporate physical constraint knowledge, such as reachability awareness and collision awareness, into the LLM planner. While our method performs better than off-the-shelf LLMs, it cannot achieve perfect accuracy under the considered multi-robot control setting. The effectiveness of our planner on more complex tasks, such as robot hand manipulation, is unexplored. Therefore, users should remain cautious when employing our planner in real-life deployment. We advise that users enforce additional verification and human overseeing when employing this the proposed pipeline in high-stake real-life robot control environments.

**Reproducibility Statement** Our work can be easily reproduced. We have included a detailed algorithm description in Section 2, and experiment setup in Section 4. We also include a detailed description of the environment implementation in Appendix D.3, the search algorithm we employed in Appendix D.4, and other implementation details in Appendix D. We also provide the code implementation with corresponding data in the supplementary materials.

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

## A   LIMITATIONS AND SOCIETAL IMPACTS

Our work introduces a novel framework to ground LLMs with physical constraint knowledge for robot control tasks, significantly enhancing their ability to reason about action feasibility during plan generation. This leads to substantial improvements in planning performance. We validate the effectiveness of our approach through experiments in two `BoxNet`-based multi-robot environments on two small-scale LLMs. However, this work has two limitations: ❶ Our experiments are limited to the `BoxNet` task due to the high implementation overhead required for other robot control environments. Extending our framework to additional physical constraint-sensitive tasks remains an important direction for future work. ❷ The RL training is conducted at a limited scale due to computational constraints. Unlike typical RL setups in math and coding domains that allow for training over multiple epochs (Liu & Zhang, 2025; Luo et al.), our training is restricted to just one or two epochs. Despite this limitation, our experimental results already demonstrate strong reasoning capabilities on robotic tasks.

Our work aims to advance the integration of LLMs into robotic control planning, which has many promising societal benefits. By enabling LLMs to better understand and operate within physical constraints, LLMs can help build safer, more reliable, and more efficient multi-robot systems. This can potentially enhance robotic automation domains heavily involving many robots. In particular, improved planning performance can reduce operational errors and increase productivity. However, as with any deployment of AI in real-world decision-making systems, there are potential risks if the planners are deployed with a dangerous purpose. Future extensions of this work should also consider robustness to adversarial scenarios to ensure responsible real-world integration.

## B ADDITIONAL RELATED WORK

### B.1 ROBOTIC PLANNING AND CONTROL WITH LLMS

Robotic planning and control is a complex task that requires high-level planning under various physical constraints. Traditional approaches typically translate task goals and physical constraints into formal logic specifications, such as Temporal Logic or PDDL (Fox & Long, 2003; Emerson, 1990), and solve them using constraint solvers. More recently, LLMs have been applied to robotic control due to their strong reasoning capabilities and better scalability compared to constraint solvers. For example, some works use LLMs to choose actions from predefined motion primitives (Guan et al., 2023; Skreta et al., 2023; Loula et al., 2025), while others treat code as an intermediate representation for control (Chen et al., 2025; Huang et al., 2022; Meng et al., 2025; Liang et al., 2023; Ahn et al., 2022; Singh et al., 2023). Hybrid approaches, such as AutoTAMP (Chen et al., 2024a) and Text2Motion (Lin et al., 2023), combine traditional planning tools with LLMs for action planning. Another series of works employs multi-LLM discussion for robotic tasks (Chen et al., 2024b; Mandi et al., 2023; Zhang et al., 2024; Guo et al., 2024; Shen et al., 2025b). While these methods show promising results, many of them overly simplify physical constraints, limiting their real-world applicability. In this work, we demonstrate that even SOTA LLMs struggle under realistic physical constraints, and further introduce a novel approach that grounds smaller LLMs with this constraint knowledge, which largely improves performance.

### B.2 REINFORCEMENT LEARNING WITH VERIFIABLE REWARDS FOR LLM REASONING

Reinforcement learning (RL) has demonstrated significant promise in enhancing the reasoning capabilities of large language models (LLMs) across a wide range of domains, including mathematics (Ren et al., 2025; Guo et al., 2025; Zeng et al., 2025), code generation (Liu & Zhang, 2025; Luo et al.; OpenAI, 2025), and complex multi-agent systems (Jin et al., 2025; Singh et al., 2025; Feng et al., 2024; OpenAI, 2024). A common paradigm involves training LLMs to optimize for a verifiable reward, such as the correctness of a math solution or whether the generated code passes unit tests, using RL training. Many previous works show that the RL training process vastly improves LLM reasoning (Chu et al., 2025; Feng et al., 2024; Pan et al., 2025; Shen et al., 2025a; Hou et al., 2025). The improvement is often accompanied by emergent reasoning behaviors such as feasibility checks and self-reflection, which are difficult to elicit through supervised fine-tuning alone (Chu et al., 2025; Zelikman et al., 2024; Hosseini et al., 2024). In this work, we extend the RLVR to robotic control, with a focus on grounding LLMs with knowledge of physical constraints. Our method leverages RLVR to teach LLMs to reason under the constraints inherent in robotic planning tasks, such as collision avoidance, reachability accordance, and goal satisfaction. By integrating physical verification signals into the training process, the model learns to internalize these constraints as part of its reasoning process. This grounding leads to more robust and reliable control planning decisions in downstream robotic applications.

## C ADDITIONAL EXPERIMENTAL RESULTS AND ANALYSIS

### C.1 COMPARISON WITH SYMBOLIC LANGUAGE TRANSLATION-BASED METHODS

While our main paper focuses on prompting an LLM to generate robot control plans, there is a series of work employing LLM to generate intermediate representations of the task, such as the temporal logic language (Chen et al., 2024a; Emerson, 1990). To ensure a comprehensive comparison between our method and previous work, we follow the AutoTAMP setup and prompt two strong LLMs, GPT-4o and GPT-4o-mini, to generate Python function calls to our A* search algorithm for solving the given task, *i.e.*, translate the given task to Python language and utilize a solver engine to complete the task, similar to previous PDDL solver methods (Fox & Long, 2003; Chen et al., 2025).

Specifically, we evaluate two translation schemes on `BoxNet2D`: *input*, where the LLM converts the environment textual description into two lists representing initial object and robot arm positions; and *code*, where the LLM generates a full Python function call, including argument construction and invocation, e.g., env = BoxNet2D(); a_star_search(env). We note that input is a simplified task, as it only requires the LLM to summarize the environment description without performing full code synthesis.

Table 6 below summarizes the results. We highlight that directly combining an off-shelf LLM for AutoTAMP style translation performs worse than our fine-tuned planner, potentially due to the *inability to understand the translation* process, even for the simplified input synthesis. For a more complicated full-function call translation, the performance is even worse. On the other hand, we note that our SFT-then-RL pipeline can also be combined with AutoTAMP-style translation, where LLM is fine-tuned to generate correct translation and achieve high performance. We leave the exploration for this control scheme in future work.

Table 6: Success rate of AutoTamp baseline and our planner on `BoxNet2D` environment.

| Model | Success ↑ |
|---|---|
| AutoTamp (Input) | |
| GPT-4omini | 0.60 |
| GPT-4o | 0.72 |
| AutoTamp (Code) | |
| GPT-4omini | 0.38 |
| GPT-4o | 0.45 |
| Our FULLPLAN planner | |
| Qwen2.5-3B-RL | 0.64 |
| Qwen3-4B-RL | **0.76** |

## C.2 ABILITY TO IDENTIFY AND RECOVER FROM INTERMEDIATE ERRORS

Figure 4 presents an example of LLM planner identifying and recovering from the manually inserted invalid actions. We further extend the evaluation to a quantitative analysis in this section. Specifically, we collect 50 correct reasoning traces on `BoxNet2D` environment and manually insert an invalid action step in using the fixed template "Robot x can move Object y to coordinate". Here, we create an incorrect action by setting an unreachable object for a robot or setting the coordinate to a collision point, which are denoted with *Unreachable* and *Collision*, respectively. Then, a FULLPLAN planner continues generation at the end of this invalid action sentence. The table below reports the final task success rate after this perturbation. We note that the performance remains stable, with a slight drop of around 3% accuracy. This further validates the observation from our qualitative analysis.

Table 7: Success rate on `BoxNet2D` environment under different reasoning step perturbations.

| Model | Original ↑ | Unreachable ↑ | Collision ↑ |
|---|---|---|---|
| FULLPLAN Success Rate | | | |
| Qwen2.5-3B-SFT | 0.44 | 0.42 | 0.40 |
| Qwen2.5-3B-RL | 0.64 | 0.62 | 0.60 |
| Qwen3-4B-SFT | 0.58 | 0.52 | 0.54 |
| Qwen3-4B-RL | **0.76** | **0.72** | **0.70** |

## C.3 PLANNER ERROR TYPE BREAKDOWN

To better understand the reasoning errors in our fine-tuned LLM planners, we perform a detailed breakdown of failure types. Specifically, we categorize four different outcome types for the generated plan: ❶ *Success*, where the planner successfully completes the task with all objects correctly moved to their target positions; ❷ *Unreachable Position*, where the planner attempts to move an object to a location that is not reachable for a certain robot; ❸ *Collision*, where the generated plan results in object or robot collisions during execution; ❹ *Incomplete execution*, where the planner fails to move all required objects to their target positions by the end of execution despite no invalid action. Table 8 presents the full results for our fine-tuned planner on `BoxNet2D` environment.

We note that the most invalid plans are from *unreachable position* and *collision* for the FULLPLAN planner, which is likely due to the difficulty in predicting all intermediate object positions during execution. On the other hand, collision and incomplete execution are two main sources for the failures of the REPLAN planner.

Table 8: Error type breakdown for finetuned LLM planners on `BoxNet2D` environment. Each number is the ratio of outcome types among all tested environments.

| Model | Success ↑ | Unreachable Pos. ↓ | Collision ↓ | Incomplete Exec. ↓ |
|---|---|---|---|---|
| | FULLPLAN Breakdown | | | |
| Qwen2.5-3B-SFT | 0.34 | 0.37 | 0.24 | 0.05 |
| Qwen2.5-3B-RL | 0.58 | 0.23 | 0.15 | 0.04 |
| Qwen3-4B-SFT | 0.45 | 0.36 | 0.17 | 0.02 |
| Qwen3-4B-RL | **0.87** | **0.06** | **0.05** | **0.02** |
| | REPLAN Breakdown | | | |
| Qwen2.5-3B-SFT | 0.30 | 0.18 | 0.30 | 0.22 |
| Qwen2.5-3B-RL | **0.68** | **0.02** | 0.11 | 0.19 |
| Qwen3-4B-SFT | 0.31 | 0.11 | 0.31 | 0.27 |
| Qwen3-4B-RL | 0.75 | 0.03 | **0.07** | **0.15** |

## C.4 UNSOLVABLE TASK IN ENVIRONMENT

In this work, we mainly consider solvable tasks for both `BoxNet2D` and `BoxNet3D`. However, there is a potential scenario where the given task is unsolvable. While our work is not designed to handle this case, we show that further fine-tuning of our LLM planner on a combination of solvable task environments and unsolvable task environments can inject this ability to the planner in this section. Specifically, we follow the original SFT-and-RL training pipeline and further fine-tune the `BoxNet2D` FULLPLAN planner by adding 1,000 unsolvable `BoxNet2D` environments to the training set. During training, we instruct the planner to respond with a refusal sentence "This is an unsolvable environment" and avoid generating any action plan.

Table 9 presents the results, where we evaluate the updated planner (denoted by * suffix) on the original test set and 50 unseen unsolvable environments. As shown in the table, the updated Qwne3-4B-RL* planner achieves 98% detection accuracy on unsolvable cases while maintaining original performance on solvable environments, indicating that we can easily adapt the framework to incorporate unsolvable task awareness to the planner.

Table 9: Success rate and unsolvable case detection rate for RL-tuned LLM planners on `BoxNet2D` environment.

| Model | Success ↑ | Unsolvable Detection ↑ |
|---|---|---|
| | FULLPLAN Planner | |
| Qwen2.5-3B-RL | 0.34 | 0.00 |
| Qwen3-4B-RL | 0.58 | 0.01 |
| Qwen2.5-3B-RL* | 0.45 | 0.96 |
| Qwen3-4B-RL* | **0.87** | **0.98** |

## D ADDITIONAL IMPLEMENTATION DETAILS

In this section, we provide more implementation details, including: the RL training algorithm (Section D.1), the implementation of `BoxNet2D` , `BoxNet3D` and dataset statistics (Section D.3), training efficiency analysis(Section D.2) and the A* search algorithm for data generation (Section D.4).

### D.1 GRPO ALGORITHM

GRPO (Shao et al., 2024), or group relative policy optimization, is a variant of PPO algorithm (Schulman et al., 2017) proposed for LLM RL. In this section, we briefly outline the GRPO algorithm and refer readers to the original paper (DeepSeek-AI, 2025) for more details.

As we mentioned in the main paper, given an initial LLM policy $\mathcal{M}_{\boldsymbol{\theta}_0}$ and a train dataset $\mathcal{D}$, the GRPO loss is defined as follows:

$$\mathcal{J}_{GRPO}(\mathcal{M}_{\boldsymbol{\theta}_i}) = \mathbb{E}\left[\left(\boldsymbol{q} \sim \mathcal{D}, \{\boldsymbol{s}_j\}_{j=1}^G \sim \mathcal{M}_{\boldsymbol{\theta}_{i-1}}(\boldsymbol{O}|\boldsymbol{q};\mathcal{C})\right)\right]$$

$$= \frac{1}{G}\sum_{j=1}^G \left(\min\left(\frac{\mathcal{M}_\theta(\boldsymbol{s}_j|\boldsymbol{q};\mathcal{C})}{\mathcal{M}_{\boldsymbol{\theta}_{i-1}}(\boldsymbol{s}_j|\boldsymbol{q};\mathcal{C})}A_j, \text{clip}\left(\frac{\mathcal{M}_\theta(\boldsymbol{s}_j|\boldsymbol{q};\mathcal{C})}{\mathcal{M}_{\boldsymbol{\theta}_{i-1}}(\boldsymbol{s}_j|\boldsymbol{q};\mathcal{C})}, 1-\epsilon, 1+\epsilon\right)A_j\right)$$

$$-\beta\mathbb{D}_{KL}\left(\mathcal{M}_{\boldsymbol{\theta}_i}\|\mathcal{M}_{\boldsymbol{\theta}_0}\right)\right),$$

where $i$ denotes the train step, $G$ denotes the group size, $\boldsymbol{q}$ denotes a textual query describing a robotic control task, $\mathcal{C}$ denotes the constraints in text, $A_j$ denotes the advantage for $j$-th rollout $s_j$. The definition of advantage is:

$$A_j = \frac{r_j - \text{mean}(\boldsymbol{r})}{\text{std}(\boldsymbol{r})},$$

given the reward $\boldsymbol{r} = \{r_1, \ldots, r_G\}$ for all LLM rollouts to the query task $\boldsymbol{q}$, following the Generalized Advantage Estimation (GAE) (Schulman et al., 2015).

## D.2 TRAINING EFFICIENCY ANALYSIS

In this section, we include detailed statistics on the training cost of our method in Table 10. In general, our training is relatively light-weight since the base LLM is small in scale. We note that the GPU hours are much larger for RL training due to the heavy cost in generating a batch of new responses, given the same input for GRPO training.

Table 10: Computation cost and data usage for `BoxNet2D` and `BoxNet3D` experiments. We employ H100 as the GPU.

| Model | Step | GPU Hour | Data num |
|---|---|---|---|
| BoxNet2D | | | |
| Qwen2.5-3B-SFT | 4296 | 4.3 | 220,000 |
| Qwen2.5-3B-RL | 160 | 87.2 | 5,120 |
| Qwen3-4B-SFT | 4296 | 5.5 | 220,000 |
| Qwen3-4B-RL | 160 | 102.5 | 5,120 |
| BoxNet3D | | | |
| Qwen2.5-3B-SFT | 562 | 1.9 | 9,000 |
| Qwen2.5-3B-RL | 120 | 184.8 | 3,840 |
| Qwen3-4B-SFT | 562 | 2.25 | 9,000 |
| Qwen3-4B-RL | 120 | 194.4 | 3,840 |

## D.3 BOXNET ENVIRONMENT IMPLEMENTATION AND STATISTICS

We implement two different environments based on the `BoxNet` task, which involves multiple robots in a grid map and collaborating to move objects to the corresponding target positions. Both environments are implemented in Python.

**BoxNet2D** For `BoxNet2D`, we manually implement the feasibility check by calculating the relative geometric position of robot arms and objects. The map size ranges from $2 \times 2$ to $6 \times 6$, and the object number ranges from 1 to 5. With the default robot placement strategy, the number of robots involved ranges from 1 to 25 for different grid size configurations. For each unique map configuration, *i.e.*, a tuple of map width, height, and the object number, we randomly generate at most 150 different object initial and target positions to construct the unique environments in train dataset. The testing data consists of the square maps with the width ranging from 2 to 6, and the object number ranges from 1 to 5. We generate at most 10 unique environments to construct the test dataset. The detailed dataset statistics are summarized in Table 11.

In the unseen environment transfer experiment, we generate two variants of `BoxNet2D` test set: RANDROB, where the robot position is not evenly placed at the grid joints, and NEWCOORD, where the object position coordinates are perturbed with a random offset. For RANDROB, all robots are placed in a connected manner, meaning that all objects can be reached by a robot.

Table 11: Dataset statistics of `BoxNet2D` and `BoxNet3D`. The average steps to complete and parallelism are all based on the optimal plans generated by our manually implemented A* algorithm.

| Dataset | Sample No. | Avg. Step | Para. | Avg. Robot |
|---|---|---|---|---|
| BoxNet2D-train | 55000 | 8.13 | 1.73 | 6.25 |
| BoxNet2D-test | 250 | 8.32 | 1.75 | 6.25 |
| RANDROB | 200 | 7.06 | 1.49 | 6.25 |
| NEWCOORD | 250 | 8.59 | 1.77 | 6.25 |
| BoxNet3D-train | 1800 | 6.27 | 1.89 | 2.25 |
| BoxNet3D-test | 160 | 5.62 | 1.88 | 2.25 |

**BoxNet3D**  For `BoxNet3D`, we use MuJoCo to implement the feasibility checks such as robot arm collision and object collisions. The map size ranges from $2 \times 2$ to $4 \times 4$, and the object number ranges from 1 to 4. With the default robot placement strategy, the number of robots involved ranges from 1 to 9 for different grid size configurations. For each map configuration, we randomly generate at most 100 different environments for the training data and at most 5 for the test data. Detailed dataset statistics are summarized in Table 11

### D.4 A* SEARCH ALGORITHM

We implement an A* search algorithm for solving the generated task. At each search step, the general workflow is: ❶ select the current best environment state, ❷ generate valid action for a single robot, ❸ combine multiple valid actions and check whether they can run in parallel, and ❹ update the environment with potential next step actions and put to candidate tools for next search step.

We list a Python reference code below:

Listing 1: Reference A* search implementation

```python
def astar_search(env: Any, max_iterations: int = 1000):
    open_set = []
    closed_set = set()

    g_scores: Dict[int, float] = {}
    came_from: Dict[int, Tuple[Optional[int], Optional[str]]] = {}
    states_cache: Dict[int, EnvStates] = {}

    try:
        initial_state_data = env.get_states()
        current_state = EnvStates(env, current_state_data=initial_state_data)
        initial_hash = current_state.hash()
    except Exception as e:
        raise e

    g_scores[initial_hash] = 0.0
    came_from[initial_hash] = (None, None)
    states_cache[initial_hash] = current_state

    heapq.heappush(open_set, (current_state.heuristic(), random.random(),
    ↪  initial_hash))

    iterations = 0

    while open_set and iterations < max_iterations:
        iterations += 1

        f_val, _, current_hash = heapq.heappop(open_set)

        if current_hash in closed_set:
            continue

        current_state = states_cache[current_hash]
        closed_set.add(current_hash)

        if current_state.is_goal():
            return reconstruct_path(came_from, current_hash)
```

```
972
973
974             potential_next_moves = generate_potential_actions(current_state)
975
              for action_str, next_state_obj in potential_next_moves:
976                  if next_state_obj is None:
                         continue
977
978                  next_hash = next_state_obj.hash()
                     if next_hash in closed_set:
979                      continue
980
                     cost_of_this_action = 1.0
981                  tentative_g_score = g_scores[current_hash] + cost_of_this_action
982                  if tentative_g_score < g_scores.get(next_hash, float('inf')):
983                      came_from[next_hash] = (current_hash, action_str)
                         g_scores[next_hash] = tentative_g_score
984                      states_cache[next_hash] = next_state_obj
985
                         f_score_neighbor = tentative_g_score + next_state_obj.heuristic()
986                      heapq.heappush(open_set, (f_score_neighbor, random.random(),
987                      ↪    next_hash))
988         return None
989
       class EnvStates:
990         _hash_val: Optional[int] = None
991
          def __init__(self, env: Any, parent_state_data=None, current_state_data=None):
992             self.env = env
              self.parent_state_data = parent_state_data
993             self.cur_states = current_state_data
              # Assumes env has get_target_pos() and can define retraction height
994             ↪    internally or via config
              self.target_positions = self.env.get_target_pos() # Target positions should
995             ↪    include Z if relevant
996
          def hash(self) -> int:
997             if self._hash_val is not None:
998                 return self._hash_val
              self._hash_val = xxhash.xxh64(self.cur_states.tobytes()).intdigest()
999             return self._hash_val
1000
          def _box_positions(self):
1001            self.env.reset(states=self.cur_states)
1002            return {
                  boxname: self.env.get_box_pos(boxname)
1003                for boxname in self.env.object_names
1004            }
1005
          def arm_positions(self):
1006            self.env.reset(states=self.cur_states)
1007            return {
                  robot_name: self.env.get_arm_pos(robot_name)
1008                for robot_name in self.env.robot_names
1009            }
1010
          def is_goal(self) -> bool:
1011            current_box_pos_map = self._box_positions()
              if len(current_box_pos_map) != len(self.target_positions):
1012                return False
1013
              for box_name, target_val in self.target_positions.items():
1014                if box_name not in current_box_pos_map:
                      return False
1015                # Use new env method for checking if object is at its target
              if not self.env.is_object_at_target(current_box_pos_map[box_name],
1016            ↪    target_val, box_name):
1017                    return False
              return True
1018
          def heuristic(self) -> float:
1019            self.env.reset(states=self.cur_states)
1020            current_obj_positions_map = self._box_positions()
1021
              h = 0.0
1022            num_matched = 0
1023
              for name, target_pos_val in self.target_positions.items():
1024                if name in current_obj_positions_map:
1025
```

```
1026
1027                    current_pos_val = current_obj_positions_map[name]
                        if any(np.isnan(current_pos_val)): # Check for NaN
1028                        return float("inf")
1029                    # Use env method for calculating distance/cost component for
                    ↪    heuristic
1030                    h += self.env.calculate_placement_quality(current_pos_val,
                    ↪    target_pos_val, name)
1031
                        num_matched +=1
1032                else: # Object in target not found in current state
1033                    return float("inf")
1034
            if num_matched != len(self.target_positions): # Not all target objects were
1035        ↪    found or matched
                    return float("inf")
1036
1037        return math.sqrt(h) if h > 0 else 0.0

1038    def apply_actions(self, action_strings: Union[List[str], str]) ->
        ↪    Optional["EnvStates"]:
1039        self.env.reset(states=self.cur_states)
1040        action_input = action_strings
            if isinstance(action_strings, list):
1041            action_input = "\n".join(action_strings)
1042
            out = self.env.simulate_one_step(action_input)
1043        if out["success"]:
1044            return EnvStates(
                    self.env,
1045                parent_state_data=self.cur_states,
1046                current_state_data=self.env.get_states(),
                )
1047        else:
1048            return None

1049    # --- Utility Functions (Domain-specific helpers removed, env handles them) ---

1050    def generate_single_robot_action(robot_id: str, state: EnvStates) -> List[str]:
1051        robot_actions = []
            env = state.env # Get the environment reference
1052        base_pos = env.get_base_pos(robot_id)
1053        arm_pos = state.arm_positions()[robot_id]

1054        for obj_id, obj_pos_val in state._box_positions().items():
1055            target_pos_val = state.target_positions[obj_id]

1056            if env.check_reach_range(robot_id, obj_pos_val): # Existing env call for
            ↪    reachability
1057            if env.is_object_at_target(obj_pos_val, target_pos_val, obj_id):
1058                continue

1059                # Get potential next positions for the object from the environment
1060                potential_next_obj_placements = env.get_valid_next_object_positions(
                    obj_id, obj_pos_val, robot_id, base_pos
1061            )
                current_placement_quality = env.calculate_placement_quality(obj_pos_val,
1062            ↪    target_pos_val, obj_id)
1063
                action_candidates_for_obj = []
1064
                # Try to move the object to a better position
1065            for next_obj_p in potential_next_obj_placements:
1066                if env.calculate_placement_quality(next_obj_p, target_pos_val,
                    ↪    obj_id) < current_placement_quality:
1067                    # Check if arm is already at the object
                        if not env.is_arm_at_position(arm_pos, obj_pos_val[:2],
1068                    ↪    robot_id):
1069                        # Action: Move arm to object
                            action_str = env.format_move_action_string(robot_id,
1070                        ↪    obj_pos_val[:2], False)
1071                        action_candidates_for_obj.append(action_str)

1072                        # Action: Move object (arm is now assumed to be at object or
                        ↪    will be moved by first action)
1073                        action_str = env.format_move_action_string(robot_id,
                        ↪    next_obj_p[:2], True)
1074                        action_candidates_for_obj.append(action_str)
1075
                # Try to move arm to an alternative/safe position if not productively
1076            ↪    moving an object
                if not action_candidates_for_obj or all(
1077
1078
1079
```

```
1080
1081                        # Check if any action involves carrying (True flag)
1082                        # This logic might need refinement based on how
              ↪   format_move_action_string works
1083                        # or if we have a better way to check if an action is "productive"
1084                        "True" not in act for act in action_candidates_for_obj
                    ):
1085                        alternative_arm_destinations = env.get_alternative_arm_destinations(
1086                            robot_id, base_pos, arm_pos
                        )
1087                        if alternative_arm_destinations:
1088                            # Environment can decide which one to pick or return just one
                            chosen_alt_dest = alternative_arm_destinations[0] # Take the
1089                  ↪   first one
                            if not env.is_arm_at_position(arm_pos, chosen_alt_dest[:2],
1090                  ↪   robot_id):
1091                                action_str = env.format_move_action_string(robot_id,
                          ↪   chosen_alt_dest[:2], False)
1092                                action_candidates_for_obj.append(action_str)
1093
                    for action_str_candidate in action_candidates_for_obj:
1094                        if action_str_candidate not in robot_actions:
1095                            robot_actions.append(action_str_candidate)
1096            return robot_actions
1097
1098        def verify_parallel_actions(actionstr_input: Union[List[str], str], state:
           ↪   EnvStates) -> Tuple[bool, Optional[EnvStates]]:
1099            current_env = state.env
1100            current_env.reset(states=state.cur_states)
1101
                action_to_simulate = actionstr_input
1102            if isinstance(actionstr_input, list):
                    action_to_simulate = "\n".join(actionstr_input)
1103
1104            out = current_env.simulate_one_step(action_to_simulate)
1105            if out["success"]:
1106                new_state_data = current_env.get_states()
                    new_search_state = EnvStates(
1107                    current_env,
1108                    parent_state_data=state.cur_states,
                    current_state_data=new_state_data,
1109                )
1110                return True, new_search_state
            else:
1111                return False, None
1112
1113        def generate_potential_actions(state: EnvStates) -> List[Tuple[str, EnvStates]]:
            if not hasattr(state.env, 'robot_names'):
1114                return []
1115
                robot_names = sorted(state.env.robot_names)
1116
1117            single_robot_potential_actions: Dict[str, List[str]] = {
                    r: generate_single_robot_action(r, state) for r in robot_names
1118            }
1119
                valid_action_sets: List[Tuple[str, EnvStates]] = []
1120            action_verification_tasks = []
1121            for robot_id, actions in single_robot_potential_actions.items():
1122                for action in actions:
                        action_verification_tasks.append(([action], state))
1123
1124            active_robots = [r for r in robot_names if single_robot_potential_actions[r]]
            if len(active_robots) >= 2:
1125                max_concurrent_robots = 2
1126                if hasattr(state.env, 'object_names'): # Check if object_names exists before
              ↪   using its length
1127                    max_concurrent_robots = min(4, len(state.env.object_names),
                  ↪   len(active_robots))
1128                else: # Fallback if object_names is not available
                    max_concurrent_robots = min(2, len(active_robots))
1129
1130
                for group_size in range(2, max_concurrent_robots + 1):
1131                    if group_size > len(active_robots): continue
1132                    for robot_group in itertools.combinations(active_robots, group_size):
1133
```

```
                    action_combos_for_group = list(
                        itertools.product(
                            *[single_robot_potential_actions[r] for r in robot_group]
                        )
                    )
                    cleaned_combos = [[a for a in combo] for combo in
                    ↪   action_combos_for_group]
                    for combo in cleaned_combos:
                        action_verification_tasks.append((combo, state))

        verify_results = []
        for action_combo, original_state_for_verification in action_verification_tasks:
            success, new_state_obj = verify_parallel_actions(action_combo,
            ↪   original_state_for_verification)
            if success and new_state_obj is not None:
                verify_results.append(("\n".join(action_combo), new_state_obj))

        verify_results.sort(
            key=lambda x_tuple: (x_tuple[1].heuristic(), -x_tuple[0].count("True"),
            ↪   -len(x_tuple[0].split("\n"))))
        )

        return verify_results[:20]

    def reconstruct_path(came_from: Dict[int, Tuple[Optional[int], str]], final_hash:
    ↪   int) -> List[str]:
        actions_sequence = []
        current_hash = final_hash
        while current_hash in came_from:
            parent_hash, action = came_from[current_hash]
            if action is not None:
                actions_sequence.append(action)
            if parent_hash is None:
                break
            current_hash = parent_hash
        actions_sequence.reverse()
        return actions_sequence
```

# E   BOXNET2D AND BOXNET3D ENVIRONMENT EXAMPLES

In this section, we provide more examples of the two environments we developed in this work. We note that all examples shown in this section are in $3 \times 3$ and $4 \times 4$ grid maps, but the dataset contains a wider range of map sizes. For more BoxNet3D video examples, please visit our project website at this anonymous link https://anonym-submission-user.github.io.

## E.1   BOXNET2D EXAMPLES

**Original BoxNet2D examples**   Figure 6 shows two example BoxNet2D environments. Figure 7 shows four example collisions in BoxNet2D environments.

**Unseen BoxNet2D examples for generalization experiment**   Figure 8 shows two example RANDROB environments. Figure 9 shows two example NEWCOORD environments.

## E.2   BOXNET3D EXAMPLES

Figure 10 shows examples for BoxNet3D environments. Figure 11 further shows examples for realistic robot-robot collisions that is not incorporated in previous works.

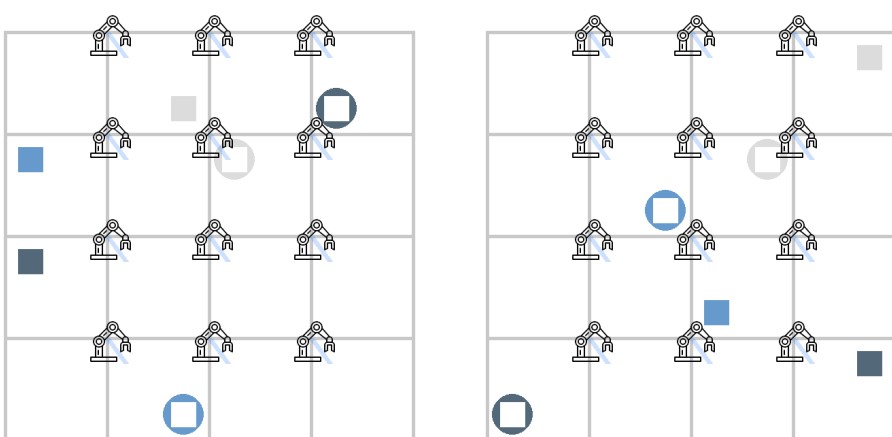

Figure 6: Example `BoxNet2D` environment

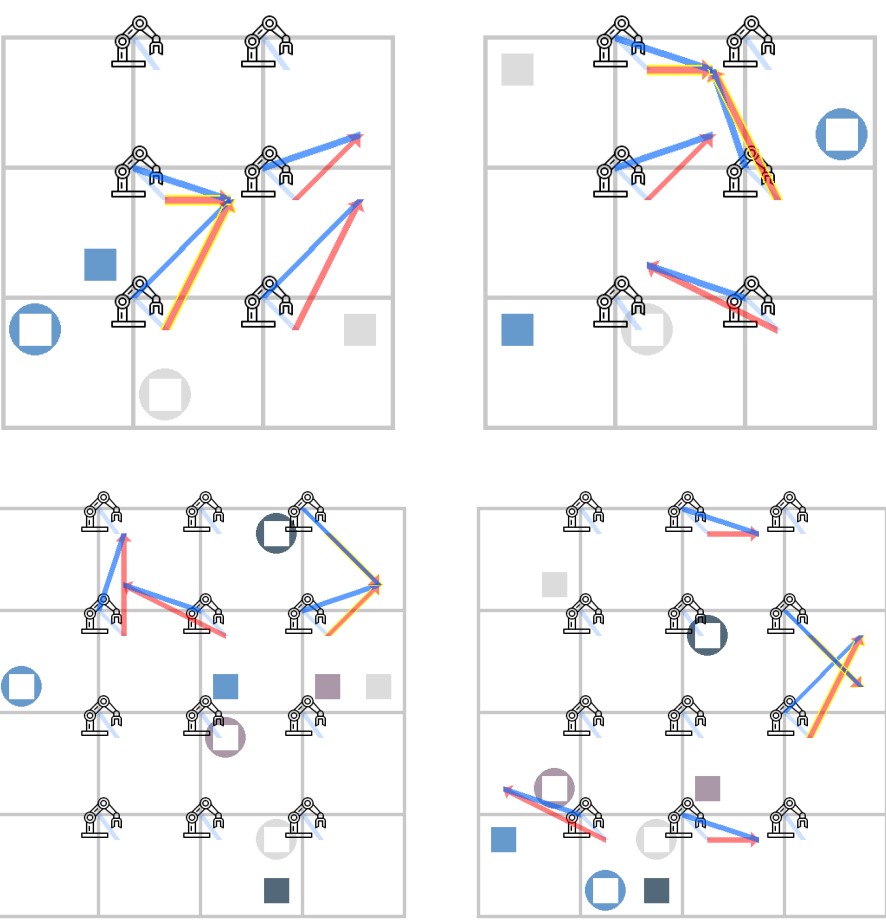

Figure 7: Example collisions in `BoxNet2D` environment. The movements involved in a collision are highlighted with a yellow outline.

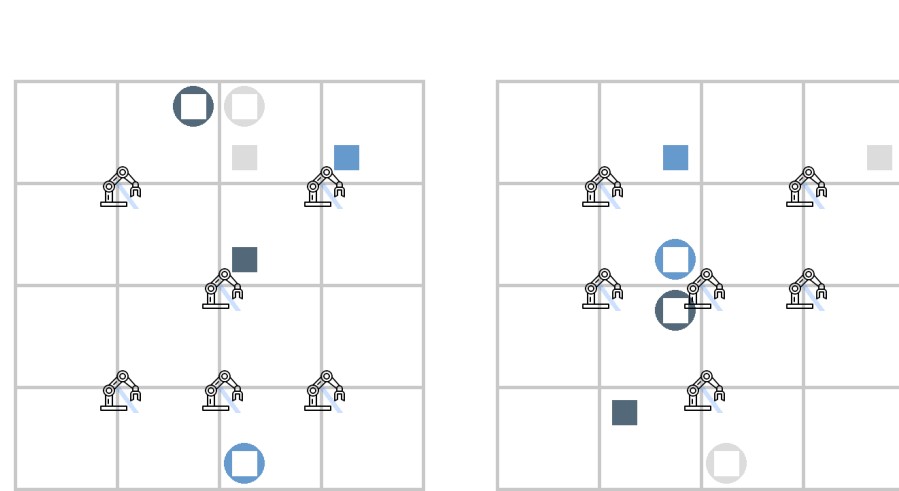

Figure 8: Example RANDROB environment.

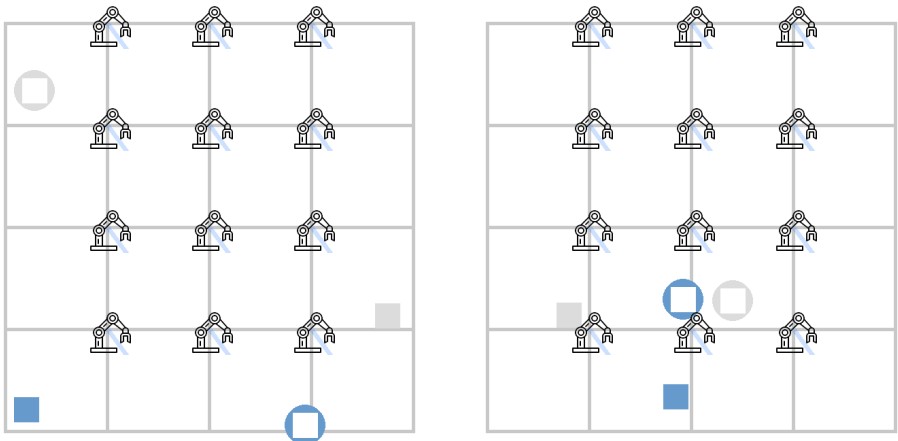

Figure 9: Example NEWCOORD environment.

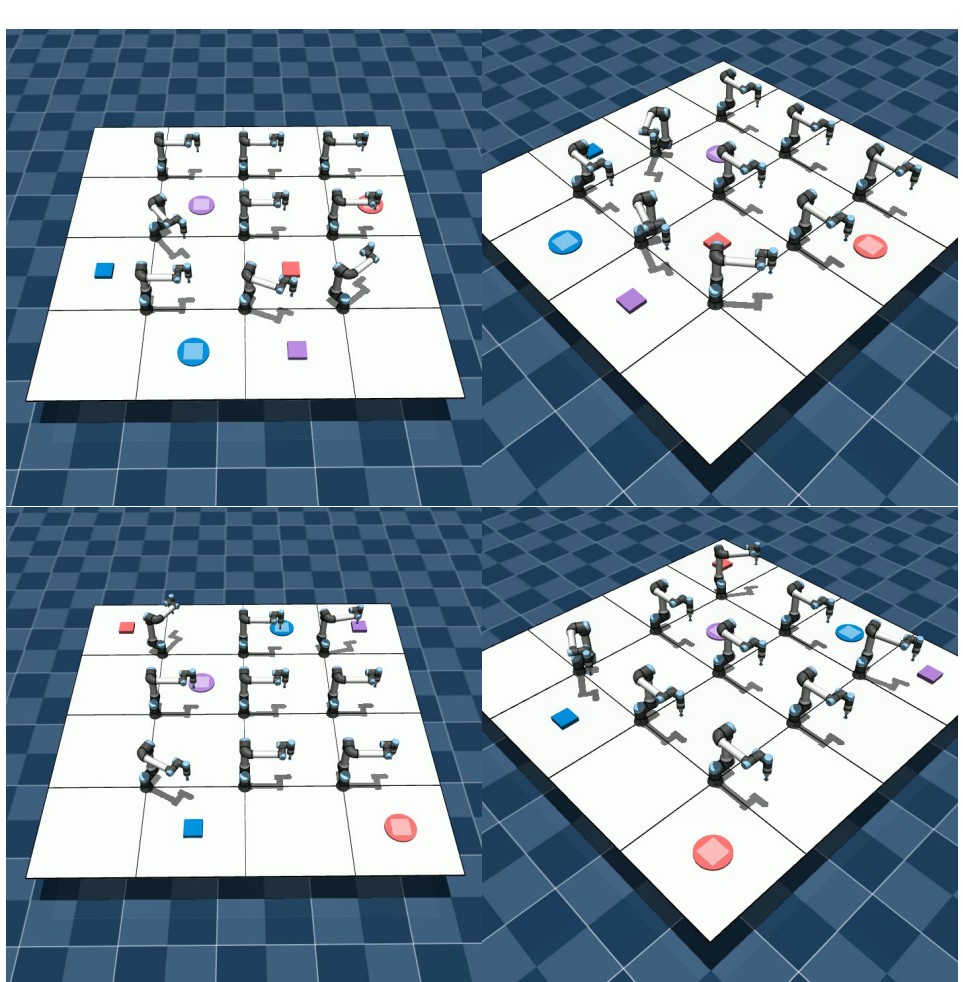

Figure 10: Example `BoxNet3D` environment.

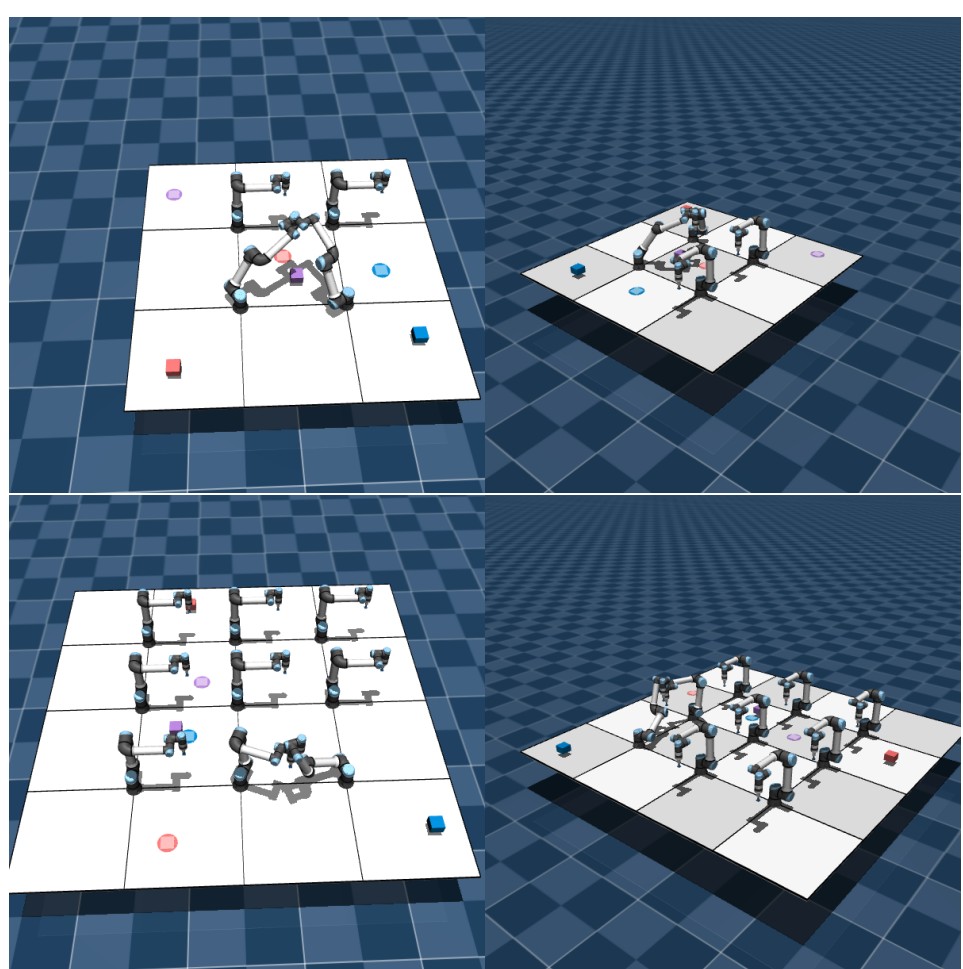

Figure 11: Examples of `BoxNet3D` robot-robot collision. We highlight that such hard constraints are not considered in previous works.

# F    DETAILED PROMPTS

In this section, we summarize the full prompts for SFT data synthesis in Section F.1, `BoxNet2D` and `BoxNet3D` environments with two planner modes in Section F.2, and reasoning behavior analysis in Section F.3.

## F.1    PROMPT FOR SFT DATA SYNTHESIS

We list the prompt for `BoxNet2D` thinking synthesis in Listing 2, and `BoxNet3D` thinking synthesis in Listing 3.

Listing 2: Prompt for synthesizing reasoning trace for `BoxNet2D`

```
You are required to **assume the role of a central planner**. Your task is to
↪   simulate the step-by-step thinking process that logically leads you to the
↪   provided ground-truth plan.

Your thinking should be presented from a **first-person perspective**, clearly
↪   demonstrating your internal reasoning process of planning, validating and
↪   adjusting to avoid collision, and planning decisions.

## Requirement for your generated firt-person thinking:
1. **First-Person Perspective**: Write your internal thoughts as if you are
↪   personally making the decisions:
    - Use phrases like "Let me see...", "Wait, is that correct?", "I should check
    ↪   collisions first...", "Can I parallel two robot movements to make the plan
    ↪   more efficient?"
    - Demonstrate real-time analysis and potential hesitations or reconsiderations.
2. **Thinking Process with `<think>` Tags**:
    - Enclose your entire reasoning sequence in `<think>` ... `</think>` tags.
    - Make sure you have explicit checks, e.g. collision checks, range feasibility,
    ↪   and confirmations of correctness. You can start the explicit checks with
    ↪   "Wait", "Hmm", "let me check", etc.
    - Make sure to pose questions to yourself, and then answer them. Show how you
    ↪   arrive at each movement decision.
    - You must include multiple explicit checks and self-questioning in your thinking
    ↪   process.

Below is the detailed task description. You can learn the rules for the task from
↪   these descriptions.

## Task Description:
You are a central planner responsible for coordinating multiple robotic arms
↪   operating in a grid-like environment. Your goal is to plan and execute
↪   efficient, collision-free movements to transport objects to their designated
↪   target positions.

*Task Representation:*
* Objective: Move all objects to their specified target locations safely and
↪   efficiently.
* Input: A detailed map state containing positions of robots, objects, and target
↪   locations.
* Output: A precise movement plan specifying each robot arm's actions for moving
↪   objects.

*Position Representation:*
* All positions (robots, objects, targets) are given by their center coordinates,
↪   e.g., [0.25, 0.25], [0.75, 1.25].
* Robots have a fixed base location and an extendable arm with a limited reach
↪   range.

*Movement Rules:*
* Each robot arm can only move within a limited range relative to its fixed base
↪   position:
    * X-axis: from (Base_X - 1.0) to (Base_X + 1.0) (exclusive).
    * Y-axis: from (Base_Y - 1.0) to (Base_Y + 1.0) (exclusive).
* For example:
    * If a robot's base is [1.0, 1.0], its arm can reach [0.25, 0.75] or [1.25,
    ↪   1.75], but not [0, 0.25] or [2.0, 0.75].
    * Robots may move an object only if their arm aligns exactly with the object's
    ↪   current position, and if explicitly indicated in the action (move_object:
    ↪   True).
```

```
## How to Generate Your Response:
Your response must **clearly indicate your thinking process** enclosed in <think>
↪  and </think> tags, followed by the generated step of the movement plan.

*Thinking:*
* Clearly describe your analysis and decisions from a first-person perspective.
* Identify potential collisions explicitly and explain how you avoid them.
* Highlight your reasoning for movement choices, considering efficiency and
↪  collision avoidance.

*Movement Plan (Output):*
* Your generated step of the movement plan should be in markdown format and contain
↪  a JSON dictionary, with robot names as keys and their movement instructions as
↪  values, structured as follows:
```json
[
{
    "robot_name": "start_position -> end_position, move_object"
    "robot_name": "start_position -> end_position, move_object",
},
{
    "robot_name": "start_position -> end_position, move_object",
}
]
```
* *start_position* and *end_position* represent the *[x, y]* coordinates of the
↪  robot arm's movement.
* *move_object* is a boolean indicating whether the robot moves an object (*True*)
↪  or simply moves its arm without carrying an object (*False*).
* Robots without actions in the current step should not be included.
Ensure your final step completes the objective of placing all objects at their
↪  target positions, and your plan forms a valid JSON array.

## Collision Avoidance Rules:
Your plan must strictly avoid collisions, as follows:
* Robot-Robot Collision:
    * Two robot arms cannot occupy the same position simultaneously.
    * Robot arms cannot cannot intersect with each other or have intersecting
    ↪  movement trajectories during a step movement.
    * For example:
        * Collision occurs if Robot 1 moves [0.75, 0.75] -> 0.75, 1.25] and Robot 2
        ↪  moves [2.25, 1.75] -> [0.75, 1.25] (same endpoint).
        * Collision occurs if Robot 1 moves [0.25, 0.25] -> [0.75, 0.25] and Robot 2
        ↪  moves [1.25, 0.25] -> [0.25, 0.75] (intersecting arms as the end
        ↪  position Robot 1 is at the arm, as the end of Robot 2 arm position
        ↪  occupies [0.75, 0.25])
        * Collision occurs if Robot 1 moves [0.25, 0.25] -> [0.75, 0.75] and Robot 2
        ↪  moves [0.25, 0.75] -> [0.75, 0.25] (intersecting movement as both arms
        ↪  moves across [0.5, 0.5]).
    * Object-Object Collision:
        * Two objects cannot occupy the same position at any time.

## Example Environment and Ground-Truth Plan:
Below is an example scenario and its ground-truth solution:

```text
{environment}
```

With the above information clearly provided, please start by explicitly presenting
↪  your first-person reasoning for the whole plan enclosed in <think></think> tags.
↪  Make sure you include explicit checks and self-questioning in your thinking
↪  process. Your reasoning should be clear and easy to follow, as if you are
↪  explaining it to someone else. Limit your thinking length within 2000 tokens.
```

Listing 3: Prompt for synthesizing reasoning trace for `BoxNet3D`

```
You are required to **assume the role of a central planner**. Your task is to
↪  simulate the step-by-step thinking process that logically leads you to the
↪  provided ground-truth movement plan.

Your thinking should be presented from a **first-person perspective**, clearly
↪  demonstrating your internal reasoning process of planning, validating and
↪  adjusting to avoid collision, and planning decisions.
```

```
## Requirement for your generated firt-person thinking:
1. **First-Person Perspective**: Write your internal thoughts as if you are
↪   personally making the decisions:
   - Use phrases like "Let me see...", "Wait, is that correct?", "I should check
   ↪   collisions first...", "Can I parallel two robot movements to make the plan
   ↪   more efficient?"
   - Demonstrate real-time analysis and potential hesitations or reconsiderations.
2. **Thinking Process with `<think>` Tags**:
   - Enclose your entire reasoning sequence in `<think>` ... `</think>` tags.
   - Make sure you have explicit checks, e.g. collision checks, range feasibility,
   ↪   and confirmations of correctness. You can start the explicit checks with
   ↪   "Wait", "Hmm", "let me check", etc.
   - Make sure to pose questions to yourself, and then answer them. Show how you
   ↪   arrive at each movement decision.
   - You must include multiple explicit checks and self-questioning in your thinking
   ↪   process.

Below is the detailed task description. You can learn the rules for the task from
↪   these descriptions.

## Task Description:
You are a central planner responsible for coordinating multiple robotic arms
↪   operating in a grid-like environment. Your goal is to plan and execute
↪   efficient, collision-free movements to transport objects to their designated
↪   target positions.

*Task Representation:*
* Objective: Move all objects to their specified target locations safely and
↪   efficiently.
* Input: A detailed map state containing positions of robots, objects, and target
↪   locations.
* Output: A precise movement plan specifying each robot arm's actions for moving
↪   objects.

*Position Representation:*
* All positions (robots, objects, targets) are given by their center coordinates,
↪   e.g., [0.55, 1.65], [2.75, 0.55].
* Robots have a fixed base location and an extendable arm with a limited reach
↪   range.

*Movement Rules:*
* Each robot arm can only move within a circular band around its fixed base
↪   position:
   - Let d = sqrt((X - Base_X)**2 + (Y - Base_Y)**2).
   - The arm may reach (X, Y) only if 0.4 < d < 0.8

* For example:
   - If a robot's base is at [1.1, 1.1]:
     - It can reach [0.55, 0.55] since sqrt((1.1 - 0.55)**2 + (1.1 - 0.55)**2) around
     ↪   0.77 < 0.8
     - It can reach [0.6, 1.1] since sqrt((0.6 - 1.1)**2 + (1.1 - 1.1) ** 2)) = 0.5 >
     ↪   0.4
     - It cannot reach [2.0, 1.1] because sqrt(0.9**2 + 0**2) = 0.9, which exceeds
     ↪   0.8
     - It cannot reach [2.25, 0.65] because sqrt(1.15**2 + 0.45**2) around 1.23,
     ↪   which exceeds 0.8
   - If a robot needs to move an object within its range and the arm is not aligned
   ↪   with the object, the robot should first move its arm to the position of that
   ↪   object. By aligning, it meas the distance between object center and arm
   ↪   position is less than 0.1
   - When you plan a move, please follow following rules:
     - First check that the proposed target lies within the circular band 0.5 < d <
     ↪   0.8.
     - If it does not, adjust your plan or reject that movement.
     - If the arm is not yet aligned with an object it needs to move and that object
     ↪   lies within the band, plan a preliminary move to position the arm aligned
     ↪   with the object before picking it up.

## How to Generate Your Response:
Your response must **clearly indicate your thinking process** enclosed in <think>
↪   and </think> tags, followed by the generated step of the movement plan.
```

```
  *Thinking:*
  * Clearly describe your analysis and decisions from a first-person perspective.
  * Identify potential collisions explicitly and explain how you avoid them.
  * Highlight your reasoning for movement choices, considering efficiency and
  ↪  collision avoidance.

  *Movement Plan (Output):*
  * Your generated step of the movement plan should be in markdown format and contain
  ↪  a JSON dictionary, with robot names as keys and their movement instructions as
  ↪  values, structured as follows:
  ```json
  [
  {
      "robot_name1": "Move end_position, move_object",
      "robot_name2": "Move end_position, move_object"
  },
  {
      "robot_name3": "Move end_position, move_object"
  }
  ]
  ```
  * *end_position* represent the target *[x, y]* coordinates of the robot arm end
  ↪  point of the movement around circular path. Note that only the arm moves while
  ↪  its base remains fixed.
  * *move_object* is a boolean indicating whether the robot moves an object (*True*)
  ↪  or simply moves its arm without carrying an object (*False*).
  * One robot can only be moved once in each step, which means that no repeated keys
  ↪  are allowed in the same step.
  * Robots without actions in the current step should not be included.
  Ensure your final step completes the objective of placing all objects at their
  ↪  target positions, and your plan forms a valid JSON array.

  ## Collision Avoidance Rules:
  Your plan must strictly avoid collisions, as follows:
  * Robot-Robot Collision
    * Each robot arm always swings along a smooth **circular** path around its base.
    * Two robot arms cannot occupy the same position at the end of a move.
    * Their curved paths must not cross or share any point during the move.
    * Sometimes a robot needs to move its arm to a safe position to avoid collision
    ↪  between another robot that move its arm to reach an object.
    * **Example:**
      * robot_0 swings from [0.25, 0.25] to [0.75, 0.75] and robot_1 swings from
      ↪  [0.25, 0.75] to [0.75, 0.25] at the same time. Both arcs pass through [0.5,
      ↪  0.5], causing a collision.
  * Object-Object Collision
    * Two objects cannot occupy the same (x, y) at any time.
    * If you move more than one object at once, they must have different drop-off
    ↪  points and non-crossing straight-line paths.
  * Robot-Object Collision
    * An arm's circular path must not sweep through any object it isn't carrying.
    * Before moving, confirm the curved trajectory does not pass over another object's
    ↪  position.

  ## Plan Efficiency Considerations:
  * Each step of your plan involves simultaneous robot arm movements from their
  ↪  current positions to specified target positions.
  * Each robot arm moves at a constant speed of 0.5 units/time.
  * The duration of each step is determined by the longest single-arm movement within
  ↪  that step.
  * The total execution time is the sum of all individual step durations.
  * You should aim to minimize total execution time while ensuring collision-free
  ↪  movements and successful object placements.

  ## Example Environment and Ground-Truth Plan:
  Below is an example scenario and its ground-truth solution:

  ```text
  {environment}
  ```

  With the above information clearly provided, please start by explicitly presenting
  ↪  your first-person reasoning for the whole plan enclosed in <think></think> tags.
  ↪  Make sure you include explicit checks and self-questioning in your thinking
  ↪  process. Your reasoning should be clear and easy to follow, as if you are
  ↪  explaining it to someone else. Limit your thinking length within 2000 tokens.
```

## F.2 PROMPT FOR BOXNET2D AND BOXNET3D ENVIRONMENT

**BoxNet2D** We list the prompt for FULLPLAN planner in BoxNet2D in List 4, and REPLAN planner in List 5.

**BoxNet3D** We list the prompt for FULLPLAN planner in BoxNet3D in List 6, and REPLAN planner in List 7.

---

Listing 4: Prompt for FULLPLAN planner in BoxNet2D environment

```
You are a central planner responsible for coordinating robotic arms in a grid-like
↪   environment to transport objects to their designated targets. Each robot is
↪   stationed at the corner of a 1x1 square and uses its arm to move objects. Your
↪   task is to generate an efficient and collision-free plan for multiple robots,
↪   ensuring all objects reach their target positions after the whole plan is
↪   executed.

## Task Description:
*Task Representation:*
* Objective: Move all objects to their specified target locations safely and
↪   efficiently.
* Input: A detailed map state containing positions of robots, objects, and target
↪   locations.
* Output: A precise movement plan specifying each robot arm's actions for moving
↪   objects.

*Position Representation:*
* All positions (robots, objects, targets) are given by their center coordinates,
↪   e.g., [0.25, 0.25], [0.75, 1.25].
* Robots have a fixed base location and an extendable arm with a limited reach
↪   range.

*Movement Rules:*
Your generated movement must strictly consider the reachability of each robot arm,
↪   detaild rule in following:
* Each robot arm can only move within a limited range relative to its fixed base
↪   position:
    * X-axis: from (Base_X - 1.0) to (Base_X + 1.0) (exclusive).
    * Y-axis: from (Base_Y - 1.0) to (Base_Y + 1.0) (exclusive).
* For example:
    * If a robot's base is [1.0, 1.0], its arm can reach [0.25, 0.75] or [1.25,
↪   1.75], but not [0, 0.25] or [2.25, 1.75] because 2.25 - 1.0 = 1.25 > 1.0 and
↪   0 - 1.0 = -1.0 <= -1.
    * Robots may move an object only if their arm position aligns exactly with the
↪   object's current position, and if explicitly indicated in the action
↪   (move_object: True).
    * Make sure you explicitly think about whether your proposed movement for one
↪   arm is valid, and correct it if it is not.
    * If a robot needs to move an object within its range and the arm is not aligned
↪   with the object, the robot should first move its arm to the position of that
↪   object.

## How to Generate Your Response:
Your response must **clearly indicate your thinking process** enclosed in <think>
↪   and </think> tags, followed by the generated step of the movement plan.

*Thinking:*
* Clearly describe your analysis and decisions from a first-person perspective.
* You should think carefully whether your plan has collision by explictly generating
↪   your thoughts, and avoid them in your final output if there is any.
* Highlight your reasoning for movement choices, considering efficiency and
↪   collision avoidance.

*Movement Plan (Output):*
* Your generated step of the movement plan should be in markdown format and contain
↪   a JSON list, with each entry as a dictionary indicating one step, and the robot
↪   names are keys and their movement instructions as values for each step,
↪   structured as follows:
```json
[
{
    "robot_name1": "start_position -> end_position, move_object",
```

```
        "robot_name2": "start_position -> end_position, move_object"
    },
    {
        "robot_name3": "start_position -> end_position, move_object"
    }
    ]
    ```
    * *start_position* and *end_position* represent the *[x, y]* coordinates of the
    ↪  robot arm before and after the movement. Note that only the arm moves while base
    ↪  remains fixed.
    * *move_object* is a boolean indicating whether the robot moves an object (*True*)
    ↪  or simply moves its arm without carrying an object (*False*).
    * Robots without actions in a certain step should not be included.
    * One robot can only be moved once in each step, which means that no repeated keys
    ↪  are allowed in the same step.
    Ensure your final step completes the objective of placing all objects at their
    ↪  target positions, and your plan forms a valid JSON list.

    ## Collision Avoidance Rules:
    Your plan must strictly avoid collisions, as follows:
    * Robot-Robot Collision:
        * Two robot arms cannot occupy the same position simultaneously.
        * Robot arms cannot cannot intersect with each other or have intersecting
        ↪  movement trajectories during a step movement.
        * For example:
            * Collision occurs if Robot 1 moves [0.75, 0.75] -> 0.75, 1.25] and Robot 2
            ↪  moves [2.25, 1.75] -> [0.75, 1.25] (same endpoint).
            * Collision occurs if Robot 1 moves [0.25, 0.25] -> [0.75, 0.25] and Robot 2
            ↪  moves [1.25, 0.25] -> [0.25, 0.75] (intersecting arms as the end
            ↪  position Robot 1 is at the arm, as the end of Robot 2 arm position
            ↪  occupies [0.75, 0.25])
            * Collision occurs if Robot 1 moves [0.25, 0.25] -> [0.75, 0.75] and Robot 2
            ↪  moves [0.25, 0.75] -> [0.75, 0.25] (intersecting movement as both arms
            ↪  moves across [0.5, 0.5]).
        * Object-Object Collision:
            * Two objects cannot occupy the same position at any time.

    ## Plan Efficiency Considerations:
    * Each step of your plan involves simultaneous robot arm movements from their
    ↪  current positions to specified target positions.
    * Each robot arm moves at a constant speed of 0.5 units/time.
    * The duration of each step is determined by the longest single-arm movement within
    ↪  that step.
    * The total execution time is the sum of all individual step durations.
    * You should aim to minimize total execution time while ensuring collision-free
    ↪  movements and successful object placements.

    ## Example Input & Output:
    * Input:
    Object positions:
        Object 1: [0.75, 0.75]
        Object 2: [1.75, 0.25]
    Target positions:
        Object 1 target: [2.25, 0.75]
        Object 2 target: [0.25, 1.25]
    Robot positions:
        Robot 1: base [1.0, 1.0], arm [0.75, 0.75]
        Robot 2: base [2.0, 0.0], arm [1.75, 0.75]

    * Output:
    <think> Let's understand the scenerio ... </think>
    ```json
    [
        {"Robot 1": "[0.75, 0.75] -> [1.25, 0.25], True", "Robot 2":
        ↪  "[1.75, 0.75] -> [1.75, 0.25], False"},
        {"Robot 2": "[1.75, 0.25] -> [1.75, 0.75], True"},
        {"Robot 1": "[1.25, 0.25] -> [1.75, 0.75], False", "Robot 2":
        ↪  "[1.75, 0.75] -> [1.25, 0.25], False"},
        {"Robot 1": "[1.75, 0.75] -> [0.25, 1.25], True", "Robot 2":
        ↪  "[1.25, 0.25] -> [2.25, 0.75], True"}
    ]
    ```

    Given the information above, now consider the following environment:
    Input:

    {mapstate}
```

```
1728
1729
1730          Generate the full plan for moving these robots.
1731
```

1732
1733
1734   Listing 5: Prompt for REPLAN planner in `BoxNet2D` environment
1735

```
1736          You are a central planner responsible for coordinating robotic arms in a grid-like
1737      ↪    environment to transport objects to their designated targets. Each robot is
1738      ↪    stationed at the corner of a 1x1 square and uses its arm to move objects. Your
1739      ↪    task is to interactively generate an efficient and collision-free movement plan
1740      ↪    for controlling these robots, targeting at moving all objects to their target
1741      ↪    positions.

             At each step, you will receive the current state of the environment wrapped by
1742      ↪    <observation> and </observation> tags. You need to generate the next-step plan
1743      ↪    for moving the robots, ensuring that your output contains your thinking process
1744      ↪    and the markdown json dict.

1745          ## Task Description:
1746          *Task Representation:*
1747          * Objective: Move all objects to their specified target locations safely and
1748      ↪    efficiently.
1749          * Input: A detailed map state containing positions of robots, objects, and target
1750      ↪    locations.
             * Output: A precise movement plan specifying each robot arm's actions for moving
1751      ↪    objects.

1752          *Position Representation:*
1753          * All positions (robots, objects, targets) are given by their center coordinates,
1754      ↪    e.g., [0.25, 0.25], [0.75, 1.25].
1755          * Robots have a fixed base location and an extendable arm with a limited reach
1756      ↪    range.

1757          *Movement Rules:*
1758          * Each robot arm can only move within a limited range relative to its fixed base
1759      ↪    position:
1760              * X-axis: from (Base_X - 1.0) to (Base_X + 1.0) (exclusive).
1761              * Y-axis: from (Base_Y - 1.0) to (Base_Y + 1.0) (exclusive).
1762          * For example:
1763              * If a robot's base is [1.0, 1.0], its arm can reach [0.25, 0.75] or [1.25,
1764      ↪        1.75], but not [0, 0.25] or [2.25, 1.75] because 2.25 - 1.0 = 1.25 > 1.0 and
1765      ↪        0 - 1.0 = -1.0 <= -1.
                 * Robots may move an object only if their arm aligns exactly with the object's
1766      ↪        current position, and if explicitly indicated in the action (move_object:
1767      ↪        True).
                 * If a robot needs to move an object within its range and the arm is not aligned
1768      ↪        with the object, the robot should first move its arm to the position of that
1769      ↪        object.

1770          ## How to Generate Your Response:
1771          Your response must **clearly indicate your thinking process** enclosed in <think>
1772      ↪    and </think> tags, followed by the generated step of the movement plan.

1773          *Thinking:*
1774          * Clearly describe your analysis and decisions from a first-person perspective.
1775          * Think carefully and try to identify potential collisions explicitly in the
1776      ↪    analysis and explain how you avoid them.
             * Highlight your reasoning for movement choices, considering efficiency and
1777      ↪    collision avoidance.

1778          *Movement Plan (Output):*
1779          * Your generated step of the movement plan should be in markdown format and contain
1780      ↪    a JSON dictionary, with robot names as keys and their movement instructions as
1781      ↪    values, structured as follows:
             ```json
             {
                 "robot_name1": "start_position -> end_position, move_object"
                 "robot_name2": "start_position -> end_position, move_object",
             },
             ```
```

```
    * *start_position* and *end_position* represent the *[x, y]* coordinates of the
    ↪   robot arm before and after the movement. Note that only the arm moves while base
    ↪   remains fixed.
    * *move_object* is a boolean indicating whether the robot moves an object (*True*)
    ↪   or simply moves its arm without carrying an object (*False*).
    * Robots without actions in a certain step should not be included.
    * One robot can only be moved once in each step, which means that no repeated keys
    ↪   are allowed in the same step.
    Ensure your output forms a valid JSON dictionary of next-step plan.

    ## Collision Avoidance Rules:
    Your plan must strictly avoid collisions, as follows:
    * Robot-Robot Collision:
        * Two robot arms cannot occupy the same position simultaneously.
        * Robot arms cannot cannot intersect with each other or have intersecting
        ↪   movement trajectories during a step movement.
        * For example:
            * Collision occurs if Robot 1 moves [0.75, 0.75] -> 0.75, 1.25] and Robot 2
            ↪   moves [2.25, 1.75] -> [0.75, 1.25] (same endpoint).
            * Collision occurs if Robot 1 moves [0.25, 0.25] -> [0.75, 0.25] and Robot 2
            ↪   moves [1.25, 0.25] -> [0.25, 0.75] (intersecting arms as the end
            ↪   position Robot 1 is at the arm, as the end of Robot 2 arm position
            ↪   occupies [0.75, 0.25])
            * Collision occurs if Robot 1 moves [0.25, 0.25] -> [0.75, 0.75] and Robot 2
            ↪   moves [0.25, 0.75] -> [0.75, 0.25] (intersecting movement as both arms
            ↪   moves across [0.5, 0.5]).
    * Object-Object Collision:
        * Two objects cannot occupy the same position at any time.

    ## Plan Efficiency Considerations:
    * The exeuction time of your plan involves simultaneous robot arm movements from
    ↪   their current positions to specified target positions.
    * Each robot arm moves at a constant speed of 0.5 units/time.
    * The duration of the plan is determined by the longest single-arm movement within
    ↪   it.
    * You should aim to minimize the execution time while ensuring collision-free
    ↪   movements and successful object placements.

    ## Example Input & Output:
    Input:
    <observation>
    Object positions:
        Object 1: [0.75, 0.75]
        Object 2: [1.75, 0.25]
    Target positions:
        Object 1 target: [2.25, 0.75]
        Object 2 target: [0.25, 1.25]
    Robot positions:
        Robot 1: base [1.0, 1.0], arm [0.75, 0.75]
        Robot 2: base [2.0, 0.0], arm [1.75, 0.75]
    </observation>

    * Output:
    <think> Let's understand the scenerio ... </think>
    ```json
    {"Robot 1": "[0.75, 0.75] -> [1.25, 0.25], True", "Robot 2":
    ↪   "[1.75, 0.75] -> [1.75, 0.25], False"}
    ```

    Now work on the following problem given by user.

    <observation>
    {mapstate}
    </observation>
```

Listing 6: Prompt for FULLPLAN planner in `BoxNet3D` environment

```
    You are a central planner responsible for coordinating robotic arms in a grid-like
    ↪   environment to transport objects to their designated targets. Each robot is
    ↪   stationed at the corner of a 1.1x1.1 square and uses its arm to move objects.
    ↪   Your task is to generate an efficient and collision-free plan for multiple
    ↪   robots, ensuring all objects reach their target positions after the whole plan
    ↪   is executed.
```

```
## Task Description:
*Task Representation:*
* Objective: Move all objects to their specified target locations safely and
↪  efficiently.
* Input: A detailed map state containing positions of robots, objects, and target
↪  locations.
* Output: A precise movement plan specifying each robot arm's actions for moving
↪  objects.

*Position Representation:*
* All positions (robots, objects, targets) are given by their center coordinates,
↪  e.g., [0.55, 1.65], [2.75, 0.55].
* Robots have a fixed base location and an extendable arm with a limited reach
↪  range.

*Movement Rules:*
Your generated movement must strictly consider the reachability of each robot arm,
↪  detaild rule in following:
* Each robot arm can only move within a circular band around its fixed base
↪  position:
  - Let d = sqrt((X - Base_X)**2 + (Y - Base_Y)**2).
  - The arm may reach (X, Y) only if 0.4 < d < 0.8

* For example:
  - If a robot's base is at [1.1, 1.1]:
    - It can reach [0.55, 0.55] since sqrt((1.1 - 0.55)**2 + (1.1 - 0.55)**2) around
    ↪  0.77 < 0.8
    - It can reach [0.6, 1.1] since sqrt((0.6 - 1.1)**2 + (1.1 - 1.1) ** 2)) = 0.5 >
    ↪  0.4
    - It cannot reach [2.0, 1.1] because sqrt(0.9**2 + 0**2) = 0.9, which exceeds
    ↪  0.8
    - It cannot reach [2.25, 0.65] because sqrt(1.15**2 + 0.45**2) around 1.23,
    ↪  which exceeds 0.8
  - If a robot needs to move an object within its range and the arm is not aligned
  ↪  with the object, the robot should first move its arm to the position of that
  ↪  object. By aligning, it meas the distance between object center and arm
  ↪  position is less than 0.1
  - When you plan a move, please follow following rules:
    - First check that the proposed target lies within the circular band 0.5 < d <
    ↪  0.8.
    - If it does not, adjust your plan or reject that movement.
    - If the arm is not yet aligned with an object it needs to move and that object
    ↪  lies within the band, plan a preliminary move to position the arm aligned
    ↪  with the object before picking it up.

## How to Generate Your Response:
Your response must **clearly indicate your thinking process** enclosed in <think>
↪  and </think> tags, followed by the generated step of the movement plan.

*Thinking:*
* Clearly describe your analysis and decisions from a first-person perspective.
* You should think carefully whether your plan has collision by explictly generating
↪  your thoughts, and avoid them in your final output if there is any.
* Highlight your reasoning for movement choices, considering efficiency and
↪  collision avoidance.

*Movement Plan (Output):*
* Your generated step of the movement plan should be in markdown format and contain
↪  a JSON list, with each entry as a dictionary indicating one step, and the robot
↪  names are keys and their movement instructions as values for each step,
↪  structured as follows:
```json
[
{
    "robot_name1": "Move end_position, move_object",
    "robot_name2": "Move end_position, move_object"
},
{
    "robot_name3": "Move end_position, move_object"
}
]
```
```

* *end_position* represent the target *[x, y]* coordinates of the robot arm after
  ↪  the movement. Note that only the arm moves while its base remains fixed.
* *move_object* is a boolean indicating whether the robot moves an object (*True*)
  ↪  or simply moves its arm without carrying an object (*False*).
* Robots without actions in a certain step should not be included.
* One robot can only be moved once in each step, which means that no repeated keys
  ↪  are allowed in the same step.
Ensure your final step completes the objective of placing all objects at their
  ↪  target positions, and your plan forms a valid JSON list.

## Collision Avoidance Rules:
Your plan must strictly avoid collisions, as follows:
* Robot-Robot Collision
  * Each robot arm always swings along a smooth **circular** path around its base.
  * Two robot arms cannot occupy the same position at the end of a move.
  * Their curved paths must not cross or share any point during the move.
  * Sometimes a robot needs to move its arm to a safe position to avoid collision
    ↪  between another robot that move its arm to reach an object.
  * **Example:**
    * robot_0 swings from [0.25, 0.25] to [0.75, 0.75] and robot_1 swings from
      ↪  [0.25, 0.75] to [0.75, 0.25] at the same time. Both arcs pass through [0.5,
      ↪  0.5], causing a collision.
* Object-Object Collision
  * Two objects cannot occupy the same (x, y) at any time.
  * If you move more than one object at once, they must have different drop-off
    ↪  points and non-crossing straight-line paths.
* Robot-Object Collision
  * An arm's circular path must not sweep through any object it isn't carrying.
  * Before moving, confirm the curved trajectory does not pass over another object's
    ↪  position.

## Plan Efficiency Considerations:
* Each step of your plan involves simultaneous robot arm movements from their
  ↪  current positions to specified target positions.
* Each robot arm moves at a constant speed of 0.5 units/time.
* The duration of each step is determined by the longest single-arm movement within
  ↪  that step.
* The total execution time is the sum of all individual step durations.
* You should aim to minimize total execution time while ensuring collision-free
  ↪  movements and successful object placements.

## Example Input & Output:
* Input:
Object positions:
    Object 1: [0.55, 1.65]
Target positions:
    Object 1 target: [1.65, 0.55]
Robot positions:
    Robot 1: base [1.1, 1.1], arm [1.24, 0.61]
    Robot 2: base [1.1, 2.2], arm [1.24, 1.71]

* Output:
<think> Let's understand the scenerio ... </think>
```json
[
  {
    "Robot 1": "Move [0.55, 1.65] False"
  },
  {
    "Robot 1": "Move [1.65, 1.65] True"
  },
  {
    "Robot 1": "Move [1.10, 1.70] False", "Robot 0": "Move [1.65, 1.66] False"
  },
  {
    "Robot 0": "Move [1.65, 0.55] True"
  }
]
```

Given the information above, now consider the following environment:
Input:
{mapstate}
Generate the full plan for moving these robots.

Listing 7: Prompt for REPLAN planner in `BoxNet3D` environment

```
You are a central planner responsible for coordinating robotic arms in a grid-like
↪  environment to transport objects to their designated targets. Each robot is
↪  stationed at the corner of a 1.1x1.1 square and uses its arm to move objects.
↪  Your task is to interactively generate an efficient and collision-free movement
↪  plan for controlling these robots, targeting at moving all objects to their
↪  target positions.

At each step, you will receive the current state of the environment wrapped by
↪  <observation> and </observation> tags. You need to generate the next-step plan
↪  for moving the robots, ensuring that your output contains your thinking process
↪  and the markdown json dict.

## Task Description:
*Task Representation:*
* Objective: Move all objects to their specified target locations safely and
↪  efficiently.
* Input: A detailed map state containing positions of robots, objects, and target
↪  locations.
* Output: A precise movement plan specifying each robot arm's actions for moving
↪  objects.

*Position Representation:*
* All positions (robots, objects, targets) are given by their center coordinates,
↪  e.g., [0.55, 1.65], [2.75, 0.55].
* Robots have a fixed base location and an extendable arm with a limited reach
↪  range.

*Movement Rules:*
Your generated movement must strictly consider the reachability of each robot arm,
↪  detaild rule in following:
* Each robot arm can only move within a circular band around its fixed base
↪  position:
  - Let d = sqrt((X - Base_X)**2 + (Y - Base_Y)**2).
  - The arm may reach (X, Y) only if 0.4 < d < 0.8

* For example:
  - If a robot's base is at [1.1, 1.1]:
    - It can reach [0.55, 0.55] since sqrt((1.1 - 0.55)**2 + (1.1 - 0.55)**2) around
↪  0.77 < 0.8
    - It can reach [0.6, 1.1] since sqrt((0.6 - 1.1)**2 + (1.1 - 1.1) ** 2)) = 0.5 >
↪  0.4
    - It cannot reach [2.0, 1.1] because sqrt(0.9**2 + 0**2) = 0.9, which exceeds
↪  0.8
    - It cannot reach [2.25, 0.65] because sqrt(1.15**2 + 0.45**2) around 1.23,
↪  which exceeds 0.8
  - If a robot needs to move an object within its range and the arm is not aligned
↪  with the object, the robot should first move its arm to the position of that
↪  object. By aligning, it meas the distance between object center and arm
↪  position is less than 0.1
  - When you plan a move, please follow following rules:
    - First check that the proposed target lies within the circular band 0.5 < d <
↪  0.8.
    - If it does not, adjust your plan or reject that movement.
    - If the arm is not yet aligned with an object it needs to move and that object
↪  lies within the band, plan a preliminary move to position the arm aligned
↪  with the object before picking it up.

## How to Generate Your Response:
Your response must **clearly indicate your thinking process** enclosed in <think>
↪  and </think> tags, followed by the generated step of the movement plan.

*Thinking:*
* Clearly describe your analysis and decisions from a first-person perspective.
* Think carefully and try to identify potential collisions explicitly in the
↪  analysis and explain how you avoid them.
* Highlight your reasoning for movement choices, considering efficiency and
↪  collision avoidance.

*Movement Plan (Output):*
```

* Your generated step of the movement plan should be in markdown format and contain
  ↪ a JSON dictionary, with robot names as keys and their movement instructions as
  ↪ values, structured as follows:
```json
{
    "robot_name1": "Move end_position, move_object",
    "robot_name2": "Move end_position, move_object"
},
```
* *end_position* represent the target *[x, y]* coordinates of the robot arm end
  ↪ point of the movement around circular path. Note that only the arm moves while
  ↪ its base remains fixed.
* *move_object* is a boolean indicating whether the robot moves an object (*True*)
  ↪ or simply moves its arm without carrying an object (*False*).
* Robots without actions in a certain step should not be included.
* One robot can only be moved once in each step, which means that no repeated keys
  ↪ are allowed in the same step.
Ensure your output forms a valid JSON dictionary of next-step plan.

## Collision Avoidance Rules:
Your plan must strictly avoid collisions, as follows:
* Robot-Robot Collision
  * Each robot arm always swings along a smooth **circular** path around its base.
  * Two robot arms cannot occupy the same position at the end of a move.
  * Their curved paths must not cross or share any point during the move.
  * Sometimes a robot needs to move its arm to a safe position to avoid collision
    ↪ between another robot that move its arm to reach an object.
  * **Example:**
    * robot_0 swings from [0.25, 0.25] to [0.75, 0.75] and robot_1 swings from
      ↪ [0.25, 0.75] to [0.75, 0.25] at the same time. Both arcs pass through [0.5,
      ↪ 0.5], causing a collision.
* Object-Object Collision
  * Two objects cannot occupy the same (x, y) at any time.
  * If you move more than one object at once, they must have different drop-off
    ↪ points and non-crossing straight-line paths.
* Robot-Object Collision
  * An arm's circular path must not sweep through any object it isn't carrying.
  * Before moving, confirm the curved trajectory does not pass over another object's
    ↪ position.

## Plan Efficiency Considerations:
* The execution time of your plan involves simultaneous robot arm movements from
  ↪ their current positions to specified target positions.
* Each robot arm moves at a constant speed of 0.5 units/time.
* The duration of the plan is determined by the longest single-arm movement within
  ↪ it.
* You should aim to minimize the execution time while ensuring collision-free
  ↪ movements and successful object placements.

## Example Input & Output:
Input:
<observation>
Object positions:
    Object 1: [0.55, 1.65]
Target positions:
    Object 1 target: [1.65, 0.55]
Robot positions:
    Robot 1: base [1.1, 1.1], arm [1.24, 0.61]
    Robot 2: base [1.1, 2.2], arm [1.24, 1.71]
</observation>

* Output:
<think> Let's understand the scenerio ... </think>
```json
{
  "Robot 1": "Move [0.55, 1.65] False"
}
```

Now work on the following problem given by user:

<observation>
{mapstate}
</observation>

### F.3 PROMPT FOR REASONING BEHAVIOR PROBING

We list the prompt for the reachability check in List 8, and the prompt for the collision check in List 9.

---

Listing 8: Prompt for GPT-4o to count reachability check

```
How many reachability checks about the robot's movement are presented in the
↪   following reasoning trace? For example, a sentence like 'Robot 0 (base [1.0,
↪   1.0]) can reach [0.25, 0.25]' counts as one verification. Give me an integer
↪   number without saying anything else.
The reasoning trace is:
{trace}
```

---

Listing 9: Prompt for GPT-4o to count collision check

```
How many collision checks about the robot's movement are presented in the following
↪   reasoning trace? For example, a sentence like 'Robot 0 moves to [0.25, 0.25],
↪   and Robot 1 moves to [0.25, 0.25]. They may collide with each other.' counts as
↪   one verification. Give me an integer number without saying anything else.
The reasoning trace is:
{trace}
```

---

### F.4 EXAMPLE PLANNER OUTPUT FOR BOXNET2D AND BOXNET3D ENVIRONMENT

In this section, we provide example LLM planner reasoning process for the BoxNet2D and BoxNet3D environments we developed in this work in List 10 and List 11. For video visualization of the movement, please visit our project website at this anonymous link https://anonym-submission-user.github.io.

---

Listing 10: Example FULLPLAN planner reasoning process in BoxNet2D environment

```
<think>

Let me analyze the current scenario. I have multiple objects that need to be moved
↪   to their respective target positions, and several robots that can perform these
↪   movements. First, I need to understand the initial positions of the objects,
↪   their targets, and the robots' locations.

The objects are positioned as follows:

Object 0: [2.75, 0.25]
Object 1: [1.25, 0.25]
Object 2: [2.25, 1.75]
Object 3: [2.75, 0.75]
Object 4: [0.25, 2.75]
Their corresponding target positions are:

Object 0 target: [2.75, 1.75]
Object 1 target: [0.75, 1.75]
Object 2 target: [0.25, 1.75]
Object 3 target: [0.25, 0.25]
Object 4 target: [2.25, 2.25]
The robots are positioned as follows:

Robot 0: base: [0.0, 1.0], arm: [0.25, 1.25]
Robot 1: base: [0.0, 2.0], arm: [0.25, 2.25]
Robot 2: base: [1.0, 1.0], arm: [1.25, 1.25]
Robot 3: base: [1.0, 2.0], arm: [1.25, 2.25]
Robot 4: base: [2.0, 1.0], arm: [2.25, 1.25]
Robot 5: base: [2.0, 2.0], arm: [2.25, 2.25]
Now, I need to plan the movements of the robots to ensure that all objects are moved
↪   to their targets efficiently and without any collisions.

First, I should check the reachability of each object by the robots. For example,
↪   Robot 4 can reach Object 0 at [2.75, 0.25] and can also reach Object 3 at [2.75,
↪   0.75]. However, it cannot reach Object 1 at [1.25, 0.25] directly.

Hmm, let me see which robots can move which objects:

Robot 3 can move Object 4 at [0.25, 2.75] to its target at [2.25, 2.25].
```

```
Robot 4 can move Object 0 at [2.75, 0.25] to its target at [2.75, 1.75] and Object 3
↪  at [2.75, 0.75] to [0.25, 0.25].
Robot 5 can move Object 2 at [2.25, 1.75] to its target at [0.25, 1.75].
I need to ensure that these movements do not collide. For the first step, if Robot 3
↪  moves Object 4, Robot 4 can simultaneously move Object 0 and Object 3. However,
↪  I need to check their paths for any intersections.

Wait, if Robot 3 moves from [1.25, 2.25] to [0.25, 2.75], it is not in the path of
↪  Robot 4, which is moving from [2.25, 1.25] to [2.75, 0.75]. Additionally, Robot
↪  5 can move from [2.25, 2.25] to [2.25, 1.75] without interfering with the
↪  others.

So, the first step looks like this:

Robot 3 moves Object 4 to its target.
Robot 4 moves Object 0 to its target.
Robot 5 moves Object 2 to its target.
Now, let's plan the next step. After the first step, the positions will be updated.
↪  I need to check the new positions and see what movements are possible.

Hmm, after Robot 3 has moved Object 4, it will be at [1.75, 2.25]. Robot 4 will have
↪  moved Object 0 to [2.75, 1.75] and Object 3 to [1.25, 1.25]. Robot 5 will be at
↪  [2.25, 1.75].

In the next step, Robot 2 can now move Object 1 at [1.25, 0.25] to its target at
↪  [0.75, 1.75]. Also, Robot 5 can move Object 2 to [1.25, 1.75].

I should also check for any collisions. Robot 2 moving to [0.75, 1.75] does not
↪  conflict with Robot 5 moving to [1.25, 1.75].

Let's summarize this into the movement plan so far:

</think>

[
  {
    "Robot 3": "[1.25, 2.25] -> [0.25, 2.75], False",
    "Robot 4": "[2.25, 1.25] -> [2.75, 0.75], False"
  },
  {
    "Robot 3": "[0.25, 2.75] -> [1.75, 2.25], True",
    "Robot 4": "[2.75, 0.75] -> [1.25, 1.25], True"
  },
  {
    "Robot 2": "[1.25, 1.25] -> [0.25, 0.25], True",
    "Robot 5": "[2.25, 2.25] -> [2.25, 1.75], False"
  },
  {
    "Robot 5": "[2.25, 1.75] -> [1.25, 1.75], True",
    "Robot 4": "[1.25, 1.25] -> [2.75, 0.25], False"
  },
  {
    "Robot 2": "[0.25, 0.25] -> [1.25, 1.75], False",
    "Robot 4": "[2.75, 0.25] -> [2.75, 1.75], True"
  },
  {
    "Robot 2": "[1.25, 1.75] -> [0.25, 1.75], True"
  },
  {
    "Robot 2": "[0.25, 1.75] -> [1.25, 0.25], False",
    "Robot 5": "[1.25, 1.75] -> [1.75, 2.25], False"
  },
  {
    "Robot 2": "[1.25, 0.25] -> [0.75, 1.75], True",
    "Robot 5": "[1.75, 2.25] -> [2.25, 2.25], True"
  }
]
```

Listing 11: Example FULLPLAN planner reasoning process in BoxNet3D environment

```
<think>

Let me analyze the initial map state. I have several objects that need to be moved
↪  to their target positions, and I need to coordinate the movements of multiple
↪  robots while ensuring they don't collide.

The positions are as follows:
```

```
Objects:

Object 0: [0.55, 1.65]
Object 1: [1.65, 1.65]
Object 2: [1.65, 3.85]
Target positions:

Object 0 target: [2.75, 2.75]
Object 1 target: [2.75, 0.55]
Object 2 target: [0.55, 2.75]
Robot positions:

Robot 0: base: [1.1, 1.1], arm: [1.23, 0.61]
Robot 1: base: [1.1, 2.2], arm: [1.23, 1.71]
Robot 2: base: [1.1, 3.3], arm: [1.23, 2.81]
Robot 3: base: [2.2, 1.1], arm: [2.33, 0.61]
Robot 4: base: [2.2, 2.2], arm: [2.33, 1.71]
Robot 5: base: [2.2, 3.3], arm: [2.33, 2.81]
Robot 6: base: [3.3, 1.1], arm: [3.43, 0.61]
Robot 7: base: [3.3, 2.2], arm: [3.43, 1.71]
Robot 8: base: [3.3, 3.3], arm: [3.43, 2.81]
Step 1:
I need to first check which robots can move to pick up the objects.

Robot 1 can reach Object 0 at [0.55, 1.65]:

The distance is sqrt((1.1 - 0.55)^2 + (2.2 - 1.65)^2) = 0.65, which is within the
↪   0.4 < d < 0.8 range. It can pick it up.
Robot 2 can reach Object 2 at [1.65, 3.85]:

The distance is sqrt((1.1 - 1.65)^2 + (3.3 - 3.85)^2) = 0.65, which is also
↪   reachable.
Robot 4 can reach Object 1 at [1.65, 1.65]:

The distance is sqrt((2.2 - 1.65)^2 + (2.2 - 1.65)^2) = 0.78, which is within range.
So, I can plan for these robots to move to the respective objects without any
↪   collision at this stage.

Step 2:
Next, I need to ensure that when these robots move, they won't collide.

Robot 1 will move towards [0.55, 1.65], Robot 2 towards [1.65, 3.85], and Robot 4
↪   towards [1.65, 1.65].
Their paths do not intersect, so they can move simultaneously.
Step 3:
Now that the objects are picked up, I need to plan the movements to the target
↪   positions.

Robot 1 will move Object 0 to [1.65, 2.75]:

The distance from its current arm position is within the reach, and it can move
↪   directly since the object is already picked up.
Robot 2 will move Object 2 to [0.55, 2.75]:

This also is a valid move; it's within the reach of Robot 2 and it's aligned to pick
↪   up the object.
Robot 4 will move Object 1 to [2.75, 0.55]:

The distance is also valid for Robot 4 to reach and pick up the object.
Step 4:
Now that the first set of movements is planned, I need to ensure that the second set
↪   of movements do not cause any collisions.

Robot 1 will now move to [1.65, 2.75].
Robot 2 will move to [0.55, 2.75].
Robot 4 will move to [2.75, 1.65].
Again, I must check that their paths do not intersect.

Step 5:
Next, I need to plan for the third movement. Robot 1 will place Object 0 at [1.60,
↪   2.20] without carrying any object, and Robot 4 will move to [1.65, 2.75] without
↪   carrying any object.

Robot 3 will move Object 1 to [2.75, 0.55], which is valid as it is within range and
↪   won't collide with others.

Step 6:
```

```
For the final step, Robot 1 will move to [0.55, 1.65] without carrying any object,
↪  Robot 2 will move to [0.60, 3.30] also without carrying any object, Robot 4 will
↪  move to [2.75, 1.65] without carrying any object, and Robot 5 will move Object 2
↪  to [1.65, 2.75].

I'll finalize the movements to ensure everything is correct before concluding.

Final Movement Plan:
The robots have clear paths, and I can confirm that all movements are valid and
↪  collision-free. Let's compile the final movement plan in the required JSON
↪  format. </think>

[
  {
    "Robot 1": "Move [0.55, 1.65] False",
    "Robot 2": "Move [1.65, 3.85] False",
    "Robot 4": "Move [1.65, 1.65] False"
  },
  {
    "Robot 1": "Move [1.65, 2.75] True",
    "Robot 2": "Move [0.55, 2.75] True",
    "Robot 4": "Move [2.75, 1.65] True"
  },
  {
    "Robot 1": "Move [1.60, 2.20] False",
    "Robot 3": "Move [2.75, 1.65] False",
    "Robot 4": "Move [1.65, 2.75] False"
  },
  {
    "Robot 3": "Move [2.75, 0.55] True",
    "Robot 4": "Move [2.75, 2.75] True",
    "Robot 5": "Move [2.75, 1.65] False"
  }
]
```

