# OpenReview forum: "Collision- and Reachability-Aware Multi-Robot Control with Grounded LLM Planners"
_ICLR.cc/2026/Conference — Submitted to ICLR 2026_

### Official Review · Reviewer_pvu6 · 2025-10-31

**Soundness:** 2
**Presentation:** 2
**Contribution:** 1
**Rating:** 2
**Confidence:** 3

**Summary:**

The paper introduces two BoxNet environments to evaluate LLM-based multi-robot control. An A* based algorithm is used to generate golden plans. Reasoning traces are collected from the A* plans and annotated by LLMs. An efficiency reward component is proposed for GRPO method. Overall the novelties of this work are not sufficient for the conference.

**Strengths:**

1. The writing is clear and the paper is easy to follow.
2. Two new environments are proposed to evaluate LLM-based multi-robot control.

**Weaknesses:**

Overall the novelties and contributions of this work are limited for the following reasons:
1. The training protocol that utilizes SFT warmup followed by GRPO is conventional. The sole modification to this standard approach is the inclusion of a simple efficiency reward term r_{efficiency} within the RL objective.

2. The proposed BoxNet2D and BoxNet3D are quite simple and straightforward. The main different from previous BoxNet is use of discrete spatial coordinates for actions instead of choosing from predefined actions.

3. The paper evaluates FULLPLAN planner and REPLAN planner but the comparison between those two settings is not fully discussed. The results of these two settings are mixed in Table 2. The claim regarding this experiment setting is unclear.

4. Errors:
Line 343: Broken reference "Figure ??"
Line 359: left "(" is missing

**Questions:**

1. Based on the formula of efficiency reward calculation, r_{efficiency} is zero when len(s) < len(s*). It penalizes the excessive length compared to the A*-based golden plan, but it won't encourage a plan with shorter length than the "golden plan". How to explain the reduced StepDiff for RL-trained model?

2. The success rate and efficiency are normally considered contradictory to each other. I would expect adding the efficient reward for RL will hurt the success rate while reducing the plan lengths. However, the results of ablation study in Table 5 shows that this efficient reward can improve the success rate while reducing the total steps. Is there any explanation or hypothesis for this phenomenon?

3. The environments have different map sizes ranging from 2x2 to 6x6 (BoxNet). How does map size affect the results of RL training?

---

> ### Author Response · Authors · 2025-11-22
>
> We thank reviewer pvu6 for the constructive comments that will help us improve this work. For your concerns, our response is detailed below:
>
> > W1: Algorithm contribution
>
> We highlight that our goal is not to propose a new RL objective in this work. Instead, we focus on integrating physical-constraint knowledge into LLM-based planners for multi-robot control tasks, enabling them to generate realistic, collision-free plans suitable for real-world execution. To the best of our knowledge, we are the first to investigate the two-stage SFT-then-RL training pipeline for such knowledge grounding.
>
> As demonstrated in our experiments, the two-stage training pipeline effectively integrates this constraint knowledge into the LLM planner. As a result, relatively small 3B/4B models outperform much larger state-of-the-art LLMs that lack such grounding (see Table 2 in the main paper).
>
> Appendix Section C.3 provides additional evidence of this physical-constraint grounding. We categorize four possible outcomes in the BoxNet2D environment: Success, Unreachable Position, Collision, and Incomplete Execution. Among these categories, Unreachable Position and Collision errors indicate the planner’s physical constraint knowledge. The results show that both collision and unreachable-position errors decrease substantially after the second-stage RL training. This confirms that our method successfully grounds physical-constraint knowledge in the LLM planner.
>
> | FullPlan breakdown | Success | Unreachable pos | Collision | Incomplete execution |
> | ------------------ | ------- | --------------- | --------- | -------------------- |
> | Qwen2.5-3B-SFT     | 0.34    | 0.37            | 0.24      | 0.05                 |
> | Qwen2.5-3B-RL      | 0.58    | 0.23            | 0.15      | 0.04                 |
> | Qwen3-4B-SFT       | 0.45    | 0.36            | 0.17      | 0.02                 |
> | Qwen3-4B-RL        | 0.87    | 0.06            | 0.05      | 0.02                 |
> |                    |         |                 |           |                      |
> | Replan breakdown   | Success | Unreachable pos | Collision | Incomplete execution |
> | Qwen2.5-3B-SFT     | 0.3     | 0.18            | 0.3       | 0.22                 |
> | Qwen2.5-3B-RL      | 0.68    | 0.02            | 0.11      | 0.19                 |
> | Qwen3-4B-SFT       | 0.31    | 0.11            | 0.31      | 0.27                 |
> | Qwen3-4B-RL        | 0.75    | 0.03            | 0.07      | 0.15                 |
>
> > W2: BoxNet2D&3D environment difference
>
> We respectfully disagree with the comment that our task environment is overly simple. Unlike prior work [1], which considers only point-overlap constraints in a 2D setting and relies on limited robot-movement actions, our environment incorporates substantially more realistic constraints. In the BoxNet2D scenario, we include movement-path intersection detection and detailed reachable-area checks. We further extend these ideas by developing a 3D simulation environment that imposes even stricter physical constraints.
>
> To the best of our knowledge, we are the first to build such a 3D simulation environment that supports scalable multi-robot control and realistic robot–robot collision detection. We provide examples of these challenging collision cases in Appendix Section E.2 of the updated PDF.
>
> [1] Chen, Yongchao, et al. "Scalable multi-robot collaboration with large language models: Centralized or decentralized systems?." 2024 IEEE International Conference on Robotics and Automation (ICRA). IEEE, 2024.
>
> > W3: FullPlan and RePlan planner comparison
>
> We would like to note that FullPlan and RePlan represent two common planning modes used in long-horizon multi-robot control applications [1]. Our goal in evaluating both modes within our two-stage training framework is not comparing which planning mode is better. Instead, we aim to demonstrate that our approach for integrating physical-constraint knowledge is generalizable across different planning paradigms.
>
> [1] Chen, Yongchao, et al. "Scalable multi-robot collaboration with large language models: Centralized or decentralized systems?." 2024 IEEE International Conference on Robotics and Automation (ICRA). IEEE, 2024.
>
> > W4: Missing reference
>
> Thanks for bringing up this missing reference, we have updated the paper pdf to fix these errors.

---

> ### Author Response · Authors · 2025-11-22
>
> > Q1: Effect of efficiency term in reward
>
> We would like to emphasize that the observation that LLMs can generate more efficient plans is consistent with prior work [1], which reports that fine-tuned transformer-based LLMs often outperform the baseline A* search used to curate their training data. This is because the A* search only provides the starting point in curating SFT data. After the SFT stage, the planner is equipped with basic planning ability. With the addition of our RL stage, the planner can further optimize plan efficiency to avoid penalty terms in the reward function.
>
> To further address your concern, we conducted an additional ablation study following the Table 5 setup, where we introduce a reward term that explicitly encourages generating plans more efficient than the A* baseline. Specifically, we remove the 0-cap, and employ following efficiency reward: $r_efficiency(s; s*) = A × (len(s) − len(s*))$, where A controls the strength of the efficiency reward, s* is the A*-generated plan, and s is the model-generated plan. Under this formulation, model-generated plans that are shorter than the A* plan receive a positive reward proportional to the step difference.
>
> We report results for A = 0.1 and A = 0.5 in the table below. Across all three variants of $r_{efficiency}$, the planner produces more efficient plans. However, the efficiency term may lower the planner's success when A becomes too large (A=0.5), potentially because overly strong efficiency reward encourages the model to prioritize shorter plans at the cost of task correctness.
>
> | Qwen2.5-RL run          | Success | StepDiff |
> | ----------------------- | ------- | -------- |
> | original r_{efficiency} | 0.58    | -0.65    |
> | A=0.1                   | 0.56    | -0.84    |
> | A=0.5                   | 0.49    | -1.13    |
>
> > Q2: Success rate&Efficiency trade-off
>
> We would like to first note that in the performance table, StepDiff is computed only over the plans that successfully solved the given task. Therefore, there is no direct contradiction between the success rate and StepDiff metrics in the reported performance table.
>
> Regarding the ablation results in Table 5, we believe the observed variations are primarily due to the common training instability issue in LLM post-training, which is largely due to the high variance in LLM rollout behaviors, as discussed in several prior works [1, 2]. Consequently, different training runs can yield noticeably different performance outcomes.
>
> To further validate this, we conducted two additional ablation RL training runs with different random seeds for the FullPlan planner without the efficiency-penalty term, following the same ablation setup described in the main paper. The results are presented in the table below. We observe that the performance variance remains high across runs. However, due to the substantial computational cost of each RL training run, we only report single run result in the main paper.
>
> | Qwen2.5-RL run | Success | StepDiff |
> | -------------- | ------- | -------- |
> | run1           | 0.52    | 1.44     |
> | run2           | 0.61    | 1.51     |
> | run3           | 0.56    | 1.37     |
> | variance       | 0.05    | 0.06     |
>
> [1] Zheng, Rui, et al. "Secrets of rlhf in large language models part i: Ppo." arXiv preprint arXiv:2307.04964 (2023).
>
> [2] Zhang, Kaichen, et al. "GVPO: Group variance policy optimization for large language model post-training." arXiv preprint arXiv:2504.19599 (2025).
>
> > Q3: Effect of the map size in training data
>
> We emphasize that our training data includes a wide range of map sizes, rom simple 2×2 grids to more complex 6×6 layouts, to ensure training diversity and enhance the planner’s generalization capability. This is reflected in the generalization experiment in Section 4.2, where we evaluate the generalization performance of the BoxNet2D planner on two previously unseen map sizes 7x7 and 10x5. We put the corresponding results in the table below. These evaluations demonstrate that the planner exhibits strong generalization, indicating that our training data is well-designed to improve planner generalization ability.
>
> |                | 7*7 grid | 10*5 grid |
> | -------------- | -------- | --------- |
> |                | success  | success   |
> | GPT-4omini     | 0.04     | 0.02      |
> | GPT-4o         | 0.08     | 0.04      |
> | Qwen2.5-3B-SFT | 0.28     | 0.22      |
> | Qwen2.5-3B-RL  | 0.36     | 0.34      |
> | Qwen3-4B-SFT   | 0.32     | 0.28      |
> | Qwen3-4B-RL    | 0.44     | 0.36      |

---

> ### Author Response · Authors · 2025-11-26
>
> Dear Reviewer pvu6,
>
> Thank you once again for your time and for providing constructive feedback on our work.
>
> We would be grateful if you could kindly take a moment to review our responses and let us know whether they sufficiently address your questions. We remain open to further discussion and clarification as needed.
>
> Thank you very much!

---

> > ### Comment · Reviewer_pvu6 · 2025-11-27
> >
> > Thanks for the response. While the proposed BoxNet2D includes "movement-path intersection detection and detailed reachable-area checks", it is more like an incremental work on top of the prior paper [1]. Given the limited contribution in the training strategy and simplified generalization setting as mentioned by reviewer uvDm, I think the overall contribution of this work is insufficient and I will keep my current rating.
> >
> > [1] Chen, Yongchao, et al. "Scalable multi-robot collaboration with large language models: Centralized or decentralized systems?." 2024 IEEE International Conference on Robotics and Automation (ICRA). IEEE, 2024.

---

> > > ### Author Response · Authors · 2025-11-27
> > >
> > > We disagree that this work is incremental. Our clarifications are as follows:
> > >
> > > 1. Our environment is not an incremental extension.
> > > [1] use a 2D discrete grid with point-overlap collision checks and simplified robot movement action. On the contrary, we introduce **path-level collision detection**, **reachable-area** checks in 2D scenario, and a newly developed **full 3D environment** with even more realistic robot–robot collision handling, none of which exist in [1].
> > >
> > > 2. We propose and validate the physical-constraint grounding training pipeline, not the environment itself. The two-stage SFT-then-RL method **explicitly teaches LLM planners physical feasibility**, leading to significant reductions in collision and unreachable-position errors. Moreover, our method leads to large empirical gains. After RL grounding, 3B/4B LLMs outperform much larger SOTA LLMs that lack physical grounding. The effectiveness of such training is not presented in [1], which is mainly prompting-based strategy for robot control.
> > >
> > > 3. The generalization challenge is not simplified. In the generalization experiment, we show that the planner generalize to **unseen map size, unseen coordinates, unseen robot layout** (Table 3 in main paper), and **unseen robot base position** (rebuttal experiment). The planner must reason about spatial relations,  collision avoidance, reachability, and long-horizon coordination. We also want to highlight that these capabilities do not emerge prompting or simple SFT training. As evidenced in Table 3, the RL-trained planner has better generalization, which is another observation not presented in previous work.
> > >
> > > Given these points, we respectfully ask the reviewer to reconsider the assessment of the contribution.

---

### Official Review · Reviewer_uvDm · 2025-11-01

**Soundness:** 2
**Presentation:** 3
**Contribution:** 1
**Rating:** 2
**Confidence:** 4

**Summary:**

This paper proposes a collision and reachability-aware framework for multi-robot control, using grounded large language model (LLM) planners. To address the inherent limitations of LLMs regarding physical and geometric reasoning, the authors introduce Reinforcement Learning with Verifiable Rewards (RLVR). RLVR fine-tunes smaller LLMs (like Qwen2.5-3B-Instruct and Qwen3-4B) with environment feedback, rewarding only physically valid plans. This method reportedly enables smaller, grounded models to outperform larger state-of-the-art models like GPT-5 in collision-free planning for up to 25 robots in the new BoxNet2D and BoxNet3D environments. The work suggests that incorporating physical constraints through RL enhances the capability and reliability of more compact LLMs for scalable robot control, though the practical scalability and computational efficiency of the RLVR process itself would benefit from further discussion.

**Strengths:**

- The main idea of the paper, grounding an LLM with physical motion constraints,  is reasonable and easy to understand. The paper is clearly written and well-structured, making the overall framework and experiments easy to follow. The claimed contribution is also clear.

- At the conceptual level, the approach to the problem seems like it could work effectively, but under structured and somewhat simpler setups.

- The code and especially the video examples clearly demonstrate the proposed framework.

**Weaknesses:**

- The core idea of grounding an LLM with physical constraints is a significant and important topic. However, the proposed approach does not seem to offer a substantial conceptual contribution, as the authors retrained a slightly different formulation of a well-known RL objective function (GRPO).

- Regarding the claimed contribution, describing the physical constraints as "realistic" seems to be an overstatement, given the simplicity of the collision and reachability checking mechanisms employed. Furthermore, the environments appear to be designed in a way that minimizes potential robot-robot collisions, which subsequently undermines the claims about reachability.

- The claimed generalization ability of this method appears to be limited. Section 4.2 describes generalization to unseen environments, but these unseen environments share the same underlying grid structure as the training environments. Consequently, the claim of generalisation slightly overstates the scope of what the proposed approach can truly handle.

- The details concerning the robot’s action space, size, and reachability conditions, as well as the collision checking method, should ideally be explained in the main body of the paper rather than being relegated to prompts in the appendix.

- The solution generation time should be considered and added as an important evaluation metric.

- The study would be significantly improved by including a comparison of the proposed RL method with other relevant works in the literature.

- Table 2 in the results section could be more self-contained and clearer. Not all the evaluation metrics are adequately defined or clear from the table caption alone.

- In the related work section, the authors argue that SoTA methods struggle due to simplified physical constraints. However, the proposed work also appears to utilize similarly simplified constraints, which creates a point of inconsistency.

- An analysis or discussion of why the A* algorithm generates a less efficient plan compared to the proposed method should be included.

- The paper should include an example output within the document itself, rather than only providing it on an external website.

- The actual use or role of RRT motion planning in the proposed framework is not clearly stated or elaborated upon.

- Figure numbers are missing in some places (e.g., Line 343).

- Missing opening parenthesis “(“ (Line 359).

- Typo: “fanalyze” should be corrected to “analyze” (Line 372).

- Sentence incomplete: The sentence starting with “due to…” (Line 862) needs completion.

**Questions:**

In addition to the issues raised above (Weaknesses), here are some further questions:

- ${A}^{\*}$ is supposed to be an optimal algorithm. I would expect any algorithm to generate a plan with the same number of steps as ${A}^{\*}$ or more. Are the heuristics used in ${A}^{\*}$ admissible?

- Why are 2D objects in BoxNet modeled as point objects?

- Why do the robots in the videos exhibit unnecessary waiting, despite the availability of a handover mechanism?

---

> ### Author Response · Authors · 2025-11-22
>
> We thank reviewer uvDm for the constructive comments that will help us improve this work. For your concerns, our response is detailed below:
>
> > W1: Contribution clarification:
>
> We highlight that our goal is not to propose a new RL objective in this work. Instead, we focus on integrating physical-constraint knowledge into LLM-based planners for multi-robot control tasks, enabling them to generate realistic, collision-free plans suitable for real-world execution. To the best of our knowledge, we are the first to investigate the two-stage SFT-then-RL training pipeline for such knowledge grounding.
>
> As demonstrated in our experiments, the two-stage training pipeline effectively integrates this constraint knowledge into the LLM planner. As a result, relatively small 3B/4B models outperform much larger state-of-the-art LLMs that lack such grounding (see Table 2 in the main paper).
>
> Appendix Section C.3 provides additional evidence of this physical-constraint grounding. We categorize four possible outcomes in the BoxNet2D environment: Success, Unreachable Position, Collision, and Incomplete Execution. Among these categories, Unreachable Position and Collision errors indicate the planner’s physical constraint knowledge. The results show that both collision and unreachable-position errors decrease substantially after the second-stage RL training. This confirms that our method successfully grounds physical-constraint knowledge in the LLM planner.
>
> | FullPlan breakdown | Success | Unreachable pos | Collision | Incomplete execution |
> | ------------------ | ------- | --------------- | --------- | -------------------- |
> | Qwen2.5-3B-SFT     | 0.34    | 0.37            | 0.24      | 0.05                 |
> | Qwen2.5-3B-RL      | 0.58    | 0.23            | 0.15      | 0.04                 |
> | Qwen3-4B-SFT       | 0.45    | 0.36            | 0.17      | 0.02                 |
> | Qwen3-4B-RL        | 0.87    | 0.06            | 0.05      | 0.02                 |
> |                    |         |                 |           |                      |
> | Replan breakdown   | Success | Unreachable pos | Collision | Incomplete execution |
> | Qwen2.5-3B-SFT     | 0.3     | 0.18            | 0.3       | 0.22                 |
> | Qwen2.5-3B-RL      | 0.68    | 0.02            | 0.11      | 0.19                 |
> | Qwen3-4B-SFT       | 0.31    | 0.11            | 0.31      | 0.27                 |
> | Qwen3-4B-RL        | 0.75    | 0.03            | 0.07      | 0.15                 |
>
> > W2: Constraint realism:
>
> We respectfully disagree with the comment that the collision and reachability checking in our task environment is overly simple. Unlike prior work [1], which considers only point-overlap constraints in a 2D setting and relies on limited robot-movement actions, our environment incorporates substantially more realistic constraints. In the BoxNet2D scenario, we include movement-path intersection detection and detailed reachable-area checks. We further extend these ideas by developing a 3D simulation environment that imposes even stricter physical constraints.
>
> To the best of our knowledge, we are the first to build such a 3D simulation environment that supports scalable multi-robot control and realistic robot–robot collision detection. We provide examples of these challenging collision cases in Appendix Section E.2 of the updated PDF.
>
> [1] Chen, Yongchao, et al. "Scalable multi-robot collaboration with large language models: Centralized or decentralized systems?." 2024 IEEE International Conference on Robotics and Automation (ICRA). IEEE, 2024.

---

> ### Author Response · Authors · 2025-11-22
>
> > W3: Planner generalization ability
>
> We acknowledge that our current generalization experiments are primarily conducted in grid-based environments. However, we emphasize that the RandRob experiment already simulates scenarios where robots are *not* uniformly distributed across the workspace. In this setting, certain grid locations are reachable only by a subset of robots, mirroring more irregular real-world layouts (see Appendix Figure 8). The RandCoord experiment further demonstrates that our planner can generalize movement actions to continuous coordinates that do not appear in the training data.
>
> To further strengthen our generalization evaluation, we additionally consider a new scenario in BoxNet2D where robots are no longer positioned at grid-joint points. Instead, some robots are placed at random points between these grid joints (Appendix Figure 10), creating more flexible robot layouts. Following the generalization experiment in the main paper, we evaluate performance on perturbed BoxNet2D test data.
>
> The results are presented in the table below. We note that our fine-tuned planners continue to exhibit robust performance, achieving up to 0.67 success for Qwen3-4B-RL fullplan planner. Consistent with our findings in the main paper, the RL-trained planners outperform the SFT-only variants, indicating stronger generalization capability.
>
> | Non-grid |                | Success | StepDiff |
> | -------- | -------------- | ------- | -------- |
> | FullPlan | Qwen2.5-3B-SFT | 0.31    | 0.31     |
> |          | Qwen2.5-3B-RL  | 0.42    | -0.12    |
> |          | Qwen3-4B-SFT   | 0.38    | 0.09     |
> |          | Qwen3-4B-RL    | 0.67    | -0.34    |
> | RePlan   | Qwen2.5-3B-SFT | 0.35    | 0.85     |
> |          | Qwen2.5-3B-RL  | 0.48    | 0.14     |
> |          | Qwen3-4B-SFT   | 0.35    | 0.17     |
> |          | Qwen3-4B-RL    | 0.61    | 0.11     |
>
> > W4: Solution generation time
>
> We agree that solution generation time is an important factor in robot control. To assess this in our task, we evaluated the FullPlan and RePlan planners on 50 BoxNet3D tasks (each in a $ 5 \times 5$ grid environment) and measured the average time required both to generate a plan and to complete the full task simulation in mujoco. This setup closely reflects real-world usage scenarios. The results are shown in the table below, along with a reference time cost for the A* search algorithm.
>
> We emphasize that the LLM-based planners are significantly more efficient than A*. The A* search requires a large number of simulations during its search process, whereas the LLM-based planners need only a few model calls, which typically complete within a few seconds.
>
> | Execution Time   |               | Solution gen (s) | Simulation (s) | Total Time (s) |
> | ---------------- | ------------- | ---------------- | -------------- | -------------- |
> |                  | A*            | -                | 308.44         | 312.41         |
> | FullPlan Planner | Qwen2.5-3B-RL | 2.39             | 2.77           | 5.16           |
> |                  | Qwen3-4B-RL   | 3.44             | 2.75           | 6.19           |
> | RePlan Planner   | Qwen2.5-3B-RL | 10.41            | 3.31           | 13.72          |
> |                  | Qwen3-4B-RL   | 13.58            | 3.22           | 16.8           |
>
> > W5: Compare with other relevant RL
>
> We want to note that our work is not focusing on proposing a new RL algorithm. Instead, we aim at integrating physical constraints knowledge to LLM-based planners in multi-robot control tasks. On the other hand, since GRPO is a representative RL algorithm and has been proven effective in many reasoning tasks like coding and math as noted in the related work section. We choose it as an implementation for incentivizing these physical constraint knowledge. We believe that applying other RL algorithms in the literature such as GSPO or DAPO does not affect the effectiveness of our framework.

---

> ### Author Response · Authors · 2025-11-22
>
> > W6: A* algorithm efficiency improvement:
>
> We would like to clarify that A* search is a heuristic-based algorithm and does not always yield optimal solutions. We use A* primarily because it significantly reduces the computational cost of exploring the large action space created by many robot candidates in the environment.
>
> In our experiments, A* serves only as a starting point for curating SFT training data. Once the SFT model acquires basic planning capability from this dataset, the subsequent RL training stage enables it to discover more efficient plans, as reflected in Table 2 performance in the main paper. This improvement is driven by the penalty term in the RL reward function, which discourages inefficient plans.
>
> This phenomenon is consistent with prior work [1], which also observes that fine-tuned transformer-based LLMs can produce more efficient plans than the basic A* algorithm originally used to construct their training data.
>
> [1] Lehnert, Lucas, et al. "Beyond a*: Better planning with transformers via search dynamics bootstrapping." arXiv preprint arXiv:2402.14083 (2024).
>
> > W7: Role of RRT motion planning
>
> We want to emphasize that our work follows prior research [1, 2], in which LLM planners generate high-level movement commands that specify target coordinates and object-movement states at each step. This abstraction simplifies task planning because the model does not need to output full robot joint configurations. However, this control paradigm requires an additional mechanism to convert each high-level movement into an executable robot joint trajectory.
>
> To bridge this gap, we adopt the same solution used in prior work [2] and integrate an RRT-based motion planner. The RRT module computes a feasible joint configuration trajectory based on the robot’s current joint state and the desired 3D end-effector target position in the workspace. This allows our system to maintain realistic, physically grounded motion execution while keeping the LLM planning high-level and manageable.
>
> [1] Chen, Yongchao, et al. "Scalable multi-robot collaboration with large language models: Centralized or decentralized systems?." 2024 IEEE International Conference on Robotics and Automation (ICRA). IEEE, 2024.
>
> [2] Mandi, Zhao, Shreeya Jain, and Shuran Song. "Roco: Dialectic multi-robot collaboration with large language models." 2024 IEEE International Conference on Robotics and Automation (ICRA). IEEE, 2024.
>
> > W8: Writing suggestion:
>
> Thanks for noting these writing issues. We have fixed them and updated the pdf file in blue texts. Specifically, we have following updates:
>
> - Action space/size/reachability condition, checking method (Section 3)
> - Example output in paper (Section F.4)
> - Table 2 caption and other typo&missing reference

---

> ### Author Response · Authors · 2025-11-22
>
> > Q1: Object is modeled as point in BoxNet2D:
>
> Following prior work [1], we model objects as points in the BoxNet2D environment, which simplifies both placement checking and collision detection. While this abstraction is convenient, we acknowledge that it is not fully realistic. For this reason, we further implement the BoxNet task in a 3D environment using the MuJoCo simulation engine. MuJoCo provides a substantially more accurate physical model, including realistic object collision geometry and detailed robot-arm motion and collision checks.
>
> Our experiments demonstrate that the proposed framework generalizes well across both the simpler 2D setting and the more complex 3D simulation environment, indicating that the overall approach is robust to more realistic physical constraints.
>
> [1] Chen, Yongchao, et al. "Scalable multi-robot collaboration with large language models: Centralized or decentralized systems?." 2024 IEEE International Conference on Robotics and Automation (ICRA). IEEE, 2024.
>
> > Q2: Video unnecessary waiting:
>
> We would like to highlight that the plans generated by our planner generally exhibit a high degree of parallelism, with multiple robots moving simultaneously to accomplish the task. In our environments, handovers are executed as follows: Robot 1 places the object at point A, after which Robot 2 moves to point A to pick up the object and continue the next step of the task.
>
> Most instances where a robot arm must wait occur because it is waiting for another robot to place the object at the handover point. These pauses are necessary to prevent robot-robot collisions and ensure safe coordination. However, we are happy to discuss further if you are referring to a different handover mechanism.

---

> ### Author Response · Authors · 2025-11-26
>
> Dear Reviewer uvDm,
>
> Thank you once again for your time and for providing constructive feedback on our work.
>
> We would be grateful if you could kindly take a moment to review our responses and let us know whether they sufficiently address your questions. We remain open to further discussion and clarification as needed.
>
> Thank you very much!

---

### Official Review · Reviewer_fabt · 2025-11-01

**Soundness:** 3
**Presentation:** 3
**Contribution:** 3
**Rating:** 6
**Confidence:** 3

**Summary:**

This paper proposes an RL‑with‑verifiable‑rewards (RLVR) pipeline to “ground” small LLMs in multi‑robot planning constraints (reachability and collision avoidance). FULLPLAN (open‑loop) and REPLAN (closed‑loop) planners are trained and evaluated. “Physical constraints” are injected via the executable reward: a plan receives credit only if it both completes the task and passes programmatic checks for reachability/feasibility and collisions; an additional efficiency penalty encourages shorter/parallelized plans. Two BoxNet environments are used: a modified 2D grid world (up to 25 robots) with analytic checks for arm reach and collisions, and a 3D MuJoCo‑based UR5e setup (up to 9 robots) with RRT‑based motion and physics‑based reachability/collision checks.

**Strengths:**

1. Clear problem focus and executable grounding. The reward integrates verifiable checks for reachability/feasibility and robot/object collisions; only physically valid, task‑completing plans are rewarded. This is a reproducible recipe for constraint‑aware planning behavior.

2. Strong empirical gains with small models. Grounded 3B/4B models outperform much larger baselines across both 2D and 3D setups.

3. Thoughtful analysis of reasoning. The paper probes for emergent feasibility checks in the chain‑of‑thought, and shows RL increases explicit reachability/collision checks in the reasoning.

4. Open‑ vs closed‑loop comparison. Evaluating FULLPLAN and REPLAN is useful for understanding where feedback helps.

5. Implementation clarity and cost accounting. The paper details datasets, constraints, prompts, GRPO settings, and compute to support reproducibility.

**Weaknesses:**

1. Prompt fairness on reachability (BoxNet2D). For BoxNet2D inference prompts, the textual context emphasizes collision rules but does not clearly encode numeric reachability limits; reachability is enforced by the simulator/reward. In contrast, BoxNet3D prompts do include explicit reachability bands/geometry. This asymmetry muddies the “prompt fairness” story across settings and may partially credit RL for implicitly learning a rule that was not textually available to zero‑shot baselines in 2D.

2. Sim‑only, single family of tasks. Results are limited to BoxNet variants; there are no real‑robot evaluations or other manipulation benchmarks, though the authors acknowledge this limitation in the paper.

3. Limited robustness stress tests. While the paper tests layout/coordinate/map variations, it does not evaluate controller noise/disturbances or perception errors, nor the latency of REPLAN vs FULLPLAN under tight timing.

4. Efficiency vs safety trade‑offs. The negative efficiency term shapes behavior (Table 5), but its sensitivity and potential side‑effects (e.g., overly aggressive parallelism near constraint boundaries) are not fully characterized.


Minor questions:
1. Would a well broken‑down reasoning process or a proven multi‑agent system help in multi‑robot tasks?

2. What exactly are the “physical constraints” incorporated into the reward? Could this be explicitly encoded into the prompts and tested with LLMs?

3. Why Qwen2.5‑3B‑Instruct and Qwen3‑4B? If one instructed model and one thinking model are preferred, I think choosing Qwen3-4B-instruct and Qwen3-4B-thinking will add one more ablation and may derive an interesting conclusion.

4. Missing figure cross-ref on Line 343.

**Questions:**

This method is simple and interesting. I have listed my concerns and some minor questions in the weakness section. Look forward to the authors' rebuttal.

---

> ### Author Response · Authors · 2025-11-22
>
> We thank reviewer fabt for the constructive comments that will help us improve this work. For your concerns, our response is detailed below:
>
>
> > W1: Box2D reachability constraint clarification:
>
> We highlight that the BoxNet2D prompt already includes detailed information about the reachable area definition in Appendix Section F.2, lines 1540–1550. This covers the reachable-area rule and an example showing how to compute the distance for the reachability check. In our work, we follow a standard prompt-design process commonly used in LLM research, with clear and well-structured prompts to describe the BoxNet task in detail for both the BoxNet2D and 3D tasks.
>
> > W2: Limited task environment:
>
> Thanks for bringing up the point about using a simulation-only environment and not including other manipulation benchmarks. To address your concern, our response is below:
>
> * Simulation only environment:
>
>      We agree that our task involves manually built simulators, which may limit its generalizability to new problems such as real-world robot control. However, this is a common challenge in many works, and we have taken the best steps we could to mitigate it by adopting a widely used task setting consistent with prior research [1]. To ensure the evaluation's reliability, we conduct experiments in two simulation environments, BoxNet2D and BoxNet3D. We believe this setting provides a meaningful benchmark for evaluating the effectiveness of our method.
>
>     Additionally, we emphasize that our focus is on high-level task planning, which is less affected by the sim-to-real gap compared to low-level control generation. As a result, while our evaluation relies on custom simulators, we believe the proposed framework remains broadly applicable if deployed for real-robot manipulation tasks.
>
> * No other manipulation benchmark
>
>     We want to highlight that our SFT-then-RL framework is general in introducing physical-constraints knowledge to the planner. This strategy has also been verified in many other reasoning domains like math and coding [2,3]. Nonetheless, we tried our best to find another reasoning task that requires heavy physical knowledge to solve the task following this work [4]. In this task, a multi-modal LLM reads an initial lego block state image and a goal state image, and outputs a high-level plan for the Lego deconstruction and construction task, such as “pick up purple, insert green.” In this process, the LLM is required to:
>
>      1. Integrate physical-constraints knowledge about which Lego block is blocking others, including support relations, interlocking geometry, and whether a block can be moved without violating clearance constraint.
>      2. Perform long-horizon task-state tracking to imagine the state changes during the long action sequence.
>     This task is closely related to our task in requiring physical-constraint knowledge, but it is operated on a single robot.
>
>     We perform a 2-stage SFT-and-RL pipeline on this task to produce the full execution plan, i.e., FullPlan planning mode in our work, using the public released data. Since the official test data is not released, we randomly sample 50 tasks from the public data for evaluation and leave the others for training. The table below shows the task success rate of the base VLM Qwen3-VL-4B-Thinking and the performance after SFT and RL. We also include the performance of several SOTA VLMs, including Gemini-2.5, GPT-5-mini, and GPT-5. All LLMs have long-reasoning capability. We highlight that the RL-trained VLM largely outperforms these SOTA VLMs, indicating that it gradually learned to reason about these physical constraints.
>
> |                      | Success rate |
> | -------------------- | ------------ |
> | Gemini-2.5           | 0.24         |
> | GPT-5-mini           | 0.16         |
> | GPT-5                | 0.38         |
> | Qwen3-VL-4B-Thinking | 0.08         |
> | Ours-SFT             | 0.56         |
> | Ours-RL              | 0.68         |
>
> [1] Chen, Yongchao, et al. "Scalable multi-robot collaboration with large language models: Centralized or decentralized systems?." 2024 IEEE International Conference on Robotics and Automation (ICRA). IEEE, 2024.
>
> [2] Zhang, Yuxiang, et al. "o1-coder: an o1 replication for coding." arXiv preprint arXiv:2412.00154 (2024).
>
> [3] Guo, Daya, et al. "Deepseek-r1: Incentivizing reasoning capability in llms via reinforcement learning." arXiv preprint arXiv:2501.12948 (2025).
>
> [4] Feng, Yunhai, et al. "Reflective planning: Vision-language models for multi-stage long-horizon robotic manipulation." arXiv preprint arXiv:2502.16707 (2025).

---

> ### Author Response · Authors · 2025-11-22
>
> > W3: Limited robustness stress test:
>
> We agree that robustness to observation is important for robot control tasks, since the real-world robot end effector position naturally has a partial offset during movement. In other words, the actual end position of the robot is offset from the planner-generated position, but the planning system should still allow tolerance to this disturbance.
>
> To further address this concern, we run a robustness stress test by injecting noise into the robot arm position at each movement step. Specifically, we randomly perturb the arm position with Normal noise in the BoxNet3D setting, i.e., $x=x+n,n∼Uniform(−A,A)$ where $A$ is the noise scale. This setup resembles real-robot execution where the controller has noise.
>
> The table below shows the success rate under different noise scales. We note that the planner performance does not degrade much when the scale is relatively small (i.e., < 0.3), indicating that our planner produces movement plans that are relatively robust to small-scale noise. However, when the noise increases, the performance gradually drops. At a scale of 1.5, the performance of all planners drops nearly to zero.
>
> | Success under different arm noise |  Model             | 0    | 0.3  | 0.5  | 1.5  |
> | --------------------------------- | ------------- | ---- | ---- | ---- | ---- |
> | FullPlan                          | Qwen2.5-3B-RL | 0.27 | 0.25 | 0.21 | 0.03 |
> |                                   | Qwen3-4B-RL   | 0.42 | 0.41 | 0.34 | 0.07 |
> | RePlan                            | Qwen2.5-3B-RL | 0.37 | 0.36 | 0.21 | 0    |
> |                                   | Qwen3-4B-RL   | 0.45 | 0.43 | 0.24 | 0    |
>
> > W4: FullPlan and RePlan latency testing
>
> We agree that latency is an important factor in robot control. To assess this in our task, we evaluated the FullPlan and RePlan planners on 50 BoxNet3D tasks (each in a $ 5 \times 5$ grid environment) and measured the average time required both to generate a plan and to complete the full task simulation in mujoco. This setup closely reflects real-world usage scenarios. The results are shown in the table below, along with a reference time cost for the A* search algorithm.
>
> We emphasize that the LLM-based planners are significantly more efficient than A*. The A* search requires a large number of simulations during its search process, whereas the LLM-based planners need only a few model calls, which typically complete within a few seconds.
>
> | Execution Time   |       Model        | Solution gen (s) | Simulation (s) | Total Time (s) |
> | ---------------- | ------------- | ---------------- | -------------- | -------------- |
> |                  | A*            | -                | 308.44         | 312.41         |
> | FullPlan Planner | Qwen2.5-3B-RL | 2.39             | 2.77           | 5.16           |
> |                  | Qwen3-4B-RL   | 3.44             | 2.75           | 6.19           |
> | RePlan Planner   | Qwen2.5-3B-RL | 10.41            | 3.31           | 13.72          |
> |                  | Qwen3-4B-RL   | 13.58            | 3.22           | 16.8           |
>
>
> > W5: Efficiency safety trader-off
>
> We agree that higher parallelism may have an impact on the execution safety since robot collision may be increased as the collision-free space becomes smaller. In practice, however, it is difficult to precisely control the level of parallelism in model-generated plans. Nonetheless, to address your concern, instead of explicitly varying the degree of parallel execution, we approximate its effect by measuring the collision error occurrence under different levels of arm noise in the W3 experiment.
>
> This injected noise simulates the motion variability that would occur when multiple robots operate, effectively reducing the safe region in a way similar to high parallelism. Using this setup, we measure the number of collisions during BoxNet3D execution under different noise scales.
>
> The results (shown in the table below) indicate that collision rates remain low when the noise is relatively small (<0.3), demonstrating a degree of robustness and preserving a usable safe region. As the noise level increases, collisions become more frequent, reflecting the inherent trade-off between efficiency and safety when scaling up parallel robot motion.
>
> | Collision count under different arm noise |  Model             | 0    | 0.3  | 0.5  | 1.5  |
> | ----------------------------------------- | ------------- | ---- | ---- | ---- | ---- |
> | FullPlan                                  | Qwen2.5-3B-RL | 0.35 | 0.37 | 0.45 | 0.51 |
> |                                           | Qwen3-4B-RL   | 0.27 | 0.31 | 0.38 | 0.47 |
> | RePlan                                    | Qwen2.5-3B-RL | 0.29 | 0.32 | 0.52 | 0.67 |
> |                                           | Qwen3-4B-RL   | 0.18 | 0.21 | 0.43 | 0.58 |

---

> ### Author Response · Authors · 2025-11-22
>
> > Q1: Breakdown-reasoning/multi-agent system:
>
> Thank you for bringing up this discussion. In our paper, our system uses a central planner to produce high-level action plans for all robots. We agree that exploring a multi-agent control system is interesting.
>
> To do this, we follow this work [1] and investigate a decentralized multi-agent system (HMAS) and compare with our central planner setting (CMAS) and use our fine-tuned RePlan planner to build a decentralized control setup. Here, the RePlan planner works as the central planner, which generates one-step actions and replans when the local planner reports invalid actions. We use GPT-4o-mini as the local planner, responsible for checking the validity of each action. Both planners work together to produce the next valid step.
>
> The table below reports the accuracy and average number of API calls in BoxNet2D setting (including both central and local planners). We note two main findings:
>
> 1) The HMAS-2 style system shows a relatively small performance gain, around 4% for both Qwen-2.5-3B-RL and Qwen-3-4B-RL planners. This is likely because the central planner was not trained to use feedback to replan.
> 2) The computational cost increases significantly, e.g., HMAS-2 API call increases from 5.6 to 24.8 for Qwen3-4B-RL. Given the small performance gain, we believe HMAS-2 may not be the best design choice for our current fine-tuned planner.
>
> | Replan mode   | CMAS success | Average API Call | HMAS success | Average API Call |
> | ------------- | ------------ | ---------------- | ------------ | ---------------- |
> | Qwen2.5-3B-RL | 0.68         | 6.1              | 0.72         | 25.5             |
> | Qwen3-4B-RL   | 0.75         | 5.6              | 0.78         | 24.8             |
>
> [1] Chen, Yongchao, et al. "Scalable multi-robot collaboration with large language models: Centralized or decentralized systems?." 2024 IEEE International Conference on Robotics and Automation (ICRA). IEEE, 2024.
>
> > Q2: Physical constraint clarification
>
> As detailed in Section 3 of the paper, the physical constraints primarily include reachability constraints and collision constraints, which are the most common sources of error in LLM-based planners. These constraints are implemented as python function calls. For instance, in BoxNet2D we use several constraint-checking functions to ensure that each robot arm remains within its reachable area and that no planned movement paths intersect. Similarly, the BoxNet3D prompt explicitly defines the reachable region and provides example rules to help the planner avoid collisions.
>
> Although these basic constraint rules are straightforward, it is challenging to fully encode all potential violation cases in the prompts. For example, specifying the exact geometric relationships involved in robot–robot collisions in Box3D task, such as the relative poses of two robot arms, is difficult to express completely in natural language. This is also why the RL training is important in improving model performance to integrate these constraints knowledge. We include additional collision examples in Section E.2 of the updated PDF.
>
>
>
> > Q3: Comparison between instruction and reasoning LLM from same base
>
>
> In our paper, we include both Qwen2.5-3B-Instruct and Qwen3-4B to cover models without strong long-reasoning abilities as well as those with such capabilities, while also covering different architectures and parameter scales (3B/4B).
>
> To further examine the effect of our training method on the same base model, we additionally apply our two-stage training procedure to Qwen3-4B-Base on the BoxNet2D task. The table below reports the results with Qwen3-4B included. We observe trends consistent with the main paper: 1. The two-stage training substantially improves planner performance. 2.The RL stage significantly enhances planning efficiency. Meanwhile, fine-tuned Qwen3-4B planners outperforms Qwen3-4B-Base variants, potentially due to stronger LLM internal reasoning capability.
>
> | BoxNet2D | Base Model        | Success | StepDiff |
> | -------- | ----------------- | ------- | -------- |
> | FullPlan | Qwen3-4B-SFT      | 0.45    | -0.12    |
> |          | Qwen3-4B-RL       | 0.87    | -0.84    |
> |          | Qwen3-4B-Base-SFT | 0.39    | 0.05     |
> |          | Qwen3-4B-Base-RL  | 0.72    | -0.47    |
> | RePlan   | Qwen3-4B-SFT      | 0.31    | -0.15    |
> |          | Qwen3-4B-RL       | 0.75    | -0.64    |
> |          | Qwen3-4B-Base-SFT | 0.28    | -0.03    |
> |          | Qwen3-4B-Base-RL  | 0.71    | -0.51    |
>
>
>
> > Q4: Missing reference:
>
> Thanks for noting the missing reference, we have updated the pdf file to fix these errors.

---

> ### Author Response · Authors · 2025-11-26
>
> Dear Reviewer fabt,
>
> Thank you once again for your time and for providing constructive feedback on our work.
>
> We would be grateful if you could kindly take a moment to review our responses and let us know whether they sufficiently address your questions. We remain open to further discussion and clarification as needed.
>
> Thank you very much!

---

### Meta-Review · Area_Chair_CP8N · 2025-12-30

**Summary:**

The primary issue from the reviewers is that the core methodology proposed in the paper is viewed as incremental rather than a significant conceptual contribution, especially compared with a well-known RL objective function, GPRO. During the rebuttal stage, the authors failed to demonstrate the novelty of the proposed method in comparison with other classic RL schemes. Further analyses of the generalization capability are missing. These concerns led two reviewers to judge the overall contribution as insufficiently novel for acceptance.

**Reviewer Concerns:**

Reviewer fabt is overall positive but less confident. However, for the remaining two reviewers (uvDm and pvu6), key concerns remain: reviewers might continue to view the methodological contribution as incremental, and the benchmarks as limited, and the generalization claims as overstated. In addition, concerns about the novelty of BoxNet still remain for Reviewer pvu6.

**Reviewer Scores:**

Since most of the concerns are not well addressed yet, I believe all the reviewers would possibly remain at the same score. So the overall score will still be 2, 2, 6.

---

### Decision · Program_Chairs · 2026-01-26

Reject